# IMPROVING GENERATIVE FLOW NETWORKS WITH PATH REGULARIZATION

## ABSTRACT

Generative Flow Networks (GFlowNets) are recently proposed models for learning stochastic policies that generate compositional objects by sequences of actions with the probability proportional to a given reward function. The central problem of GFlowNets is to improve their exploration and generalization. In this work, we propose a novel path regularization method based on optimal transport theory that places prior constraints on the underlying structure of the GFlowNet. The prior is designed to help the GFlowNet better discover the latent structure of the target distribution or enhance its ability to explore the environment in the context of active learning. The path regularization controls the flow in the GFlowNet to generate more diverse and novel candidates via maximizing the optimal transport distances between two forward policies or to improve the generalization via minimizing the optimal transport distances. In addition, we derive an efficient implementation of the regularization by finding its closed-form solutions in specific cases and a meaningful upper bound that can be used as an approximation to minimize the regularization term. We empirically demonstrate the advantages of our path regularization on a wide range of tasks, including synthetic hypergrid environment modeling, discrete probabilistic modeling, and biological sequence design.

## 1 INTRODUCTION

Recently proposed by Bengio et al. (2021a), Generative Flow Networks (GFlowNets) are generative models for compositional objects, which learn a stochastic policy that sequentially modifies a temporarily constructed object through a sequence of actions to make the generating likelihood proportional to a given reward function. Specifically, GFlowNets aim to solve the problem of generating a diverse set of good candidates. In biological sequence design, diversity is a crucial consideration because of improving the chance of discovering candidates that can satisfy many evaluation criteria later in downstream phases (Jain et al. (2022)). Especially in the multi-round active learning setting, where the generator was iteratively improved by receiving feedback from an oracle on their proposed candidates, the effect of diverse generation becomes apparent because more diversity means more exploration and knowledge gained. Besides, the generalization ability of GFlowNets (Zhang et al. (2022); Malkin et al. (2022)) over structured data makes them a good framework for discrete probabilistic modeling.

The central problems of GFlowNets are improving exploration and generalization. In this work, we propose to *train the GFlowNet with an additional path regularization via optimal transport* (Villani, 2003)), which acts as a prior constraint on its underlying structure. The prior is designed to help the GFlowNet better discover the latent structure of the target distribution or enhance its ability to explore the environment in the context of active learning. Precisely, the path regularization via OT can help the GFlowNet generate more diverse and novel candidates via maximizing OT distances between two forward policies or improving generalization via minimizing the OT distances.

**For generalization:** To improve GFlowNet's generalization, we propose the following prior constraints: (i) The forward policies of two neighbor states are expected to be similar in the way that they both have the focused tendency of choosing the next action, which implicitly forces the GFlowNet to find states with high rewards rather than exploring, especially in sparse environments. ; (ii) Trajectories related to positive objects (both have high rewards) must share their paths. As a result, the similarity of states along trajectories with high flow is higher than in other places. From a probabilistic perspective, we propose to measure the similarity of states $s$ and $s'$ in the GFlowNet by the

transition probability from $s$ to $s'$ ; (iii) When the GFlowNet learns something, the sparse flow is expected to generalize better. Thus, although many solutions exist for learning a GFlowNet, our proposal priors promote refining the GFlowNet's flow, i.e., enhance flow on high flow trajectories and vice versa.

**For diversity and exploration:** To encourage the GFlowNet's policy to generate more diverse candidates, such as in the multi-round active learning settings, we propose to put a prior constraint on the forward policies of two neighbor states. Specifically, this prior constraint intentionally promotes the "dissimilarity" between the forward policies of two neighbor states. In other words, it forces the children states of two considered neighbor states far from each other in terms of probabilistic transition, which will help the GFlowNet generate more diverse and novel candidates.

**Why OT is a good solution?** Indeed, we need a measure of "distance" between pairs of probability distributions. The optimal transport (OT) theory (Villani (2003)) studies how probabilistic mass can be optimally transported from the supports of one probabilistic distribution to the supports of another distribution given a cost function. The minimum transportation cost, called distance, can be used as a metric that quantifies the distance between two probability distributions. In the context of GFlowNets, we want to affect nearby states, which can be done by regularizing on the OT distance between the forward policies $P_F(\cdot|s)$ and $P_F(\cdot|s')$ of two neighbor states $s$ and $s'$. To compute the OT distance, we solve an OT problem between two discrete probability measures, whose support points are the child states of $s$ and $s'$ correspondingly, given the transportation cost $c(u_i, v_j)$ from each child $u_i$ of $s$ to each child $v_j$ of $s'$. While the weakness of KL divergence is that it requires two interested distributions to share the same set of supports, OT can deal with this problem efficiently. Another reason is that the cost used in our OT distance can capture the given DAG's structure and the GFlowNet's flow, while directly using KL divergence cannot.

**Contributions.** In this work, we develop a novel path regularization based on OT theory for either helping the GFlowNet better discover the latent structure of the target distribution or enhancing its ability to explore the environment in the context of active learning. Our contributions can be summarized as follows:

1. We propose to train the GFlowNet with an additional path regularization via OT, which acts as a prior on the underlying structure of the GFlowNets for either improving the generalization capability or enhancing the exploration ability of the GFlowNet.

2. We define a new directed distance between two arbitrary states in the GFlowNet, which can be naturally chosen as the transportation cost for computing the OT distance and link the proposed regularization to entropy terms.

3. We also derive an efficient implementation of the proposed regularization by finding its closed-form solutions in specific cases and a meaningful upper bound that can be used as an approximation when we want to minimize the regularization term.

**Organization.** The paper is organized as follows. In Section 2, we provide the background of GFlowNets and OT. In Section 3, we propose a new directed distance between two arbitrary states in the GFlowNet and then derive the formulation of path regularization via OT. We also explain why it is the natural and optimal choice for constructing the transportation cost between states. Then, we derive the upper bound and efficient implementation of the proposed path regularization. We provide extensive experiment results of our path regularization via OT in Section 4 and conclude the paper with a few discussions in Section 5. Theoretical proofs, as well as experimental settings and additional results, are provided in the Appendix.

## 2 BACKGROUND

### 2.1 GFLOWNETS

Given a *compositional* space $\mathcal{X}$, where each object $x \in \mathcal{X}$ can be constructed by taking a sequence of discrete *actions* from the action space $\mathcal{A}$. Specifically, the construct of each object begins from the *source state* $s_0$ and ends in the *final state* $s_f$. Incrementally, the generation process modifies a temporarily constructed object, which is called a *state* $s \in \mathcal{S}$. In addition, a specific action determines that the object is completely constructed and represents a *terminal state*, such that $s = x \in \mathcal{X}$. These states and actions correspond to the vertices and edges of a directed acyclic graph $G = (\mathcal{S}, \mathcal{A})$. The construction of an object $x \in \mathcal{X}$ defines a *complete trajectory*, which is a sequence of transitions

$\tau = (s_0 \to s_1 \to \ldots \to s_n = x \to s_f)$. $\mathcal{T}$ is defined as the set of all complete trajectories. Following Malkin et al. (2022), we may assume that each terminal state $s \in \mathcal{X}$ has only one outgoing edge, which is $s \to s_f$.

**Flows** Following Bengio et al. (2021b), a *trajectory flow* is a nonnegative function $F : \mathcal{T} \mapsto \mathbb{R}^+$, which represents the probability mass of each complete trajectory $\tau$. Consequently, the flow through each state can be defined as $F(s) = \sum_{\tau \in \mathcal{T}, s \in \tau} F(\tau)$, as well as the flow through each edge $F(s \to s') = \sum_{\tau \in \mathcal{T}, s \to s' \in \tau} F(\tau)$. We can associate a probability measure $P$ with the trajectory flow $F$. In which, there are two important conditional probabilities, the forward transition probabilities (forward policy) $P_F(s'|s) := F(s \to s')/F(s)$ is related to adding an element to build the objects, and the backward transition probabilities (backward policy) $P_B(s|s') := F(s \to s')/F(s')$ is related to removing an element.

**Learning Objective** Theoretically, if the training objective such as flow matching objective (Bengio et al., 2021a), detail balance objective (Bengio et al., 2021b), or trajectory balance objective (Malkin et al., 2022) is achieved on all states and possible trajectories respectively, then a GFlowNet can be trained to completion, i.e., perfectly generating objects proportional to their rewards. In this paper, we use trajectory balance objective because it brings more efficient credit assignment and faster convergence (Malkin et al. (2022)). We provide more background of GFlowNets in Appendix B.

## 2.2 Optimal Transport Distance

**Transportation plans and joint probabilities** For two discrete probability measures $\boldsymbol{\alpha}$ and $\boldsymbol{\beta}$ over some space $\mathcal{X}$, the admissible couplings set, which can be interpreted as the set of transportation plans or joint probability distributions, is defined as:

$$\Pi(\boldsymbol{\alpha}, \boldsymbol{\beta}) = \left\{ \pi \in \mathbb{R}_+^{k \times l} : \pi \mathbb{1}_l = \boldsymbol{\alpha}, \pi^\top \mathbb{1}_k = \boldsymbol{\beta} \right\}. \tag{1}$$

**Optimal transportation** The Kantorovich optimal transport (Peyré & Cuturi (2019)) between $\boldsymbol{\alpha}$ and $\boldsymbol{\beta}$ is defined as follows:

$$\text{OT}_\mathbf{C}(\boldsymbol{\alpha}, \boldsymbol{\beta}) := \min_{\pi \in \Pi(\boldsymbol{\alpha}, \boldsymbol{\beta})} \langle \mathbf{C}, \pi \rangle \tag{2}$$

where $\mathbf{C}$ is the cost matrix and $\mathbf{C}_{ij}$ describes the cost of transport mass from the support $i^{th}$ of $\boldsymbol{\alpha}$ toward the support $j^{th}$ of $\boldsymbol{\beta}$. Whenever the matrix $\mathbf{C}$ is itself a metric matrix, the optimum of this problem, $\text{OT}_\mathbf{C}(\boldsymbol{\alpha}, \boldsymbol{\beta})$, can be proved to be also a distance. Assuming that $k = l = d$, the worst-case complexity of computing that optimum with any of the algorithms known so far scales in $O(d^3 \log d)$ and turns out to be super-cubic in practice (Pele & Werman (2009), §2.1).

## 3 Path Regularization via Optimal Transport

### 3.1 Optimal Transport Formulation of the Path Regularization

**Turn transition probability into directed distance** We define a new directed distance between two arbitrary states in the GFlowNet, which is used as transportation cost to compute OT distance. Specifically, the directed distance from a state $s$ to another state $s'$ is designed to be inversely proportional to the probability of going from $s$ to $s'$. And, because we consider two arbitrary states $s$ and $s'$, it is trivial that there does not always exist a sequence of forward transitions $\tau$ from $s$ to $s'$ such as $\tau = (s = s_0 \to s_1 \to \ldots \to s_n = s')$ where $s_t \to s_{t+1} \in \mathcal{A}$. This will result in many entries of the cost matrix may have infinite values, which can also make the OT cost infinite. Therefore, we consider a generalized notion of $\tau$ as a sequence of transitions $\tau = (s = s_0 \to s_1 \to \ldots \to s_n = s')$ where $s_t \to s_{t+1}$ can be a forward or backward transition, i.e $\tau$ can be a back-and-forth trajectory (Zhang et al. (2022)). In fact, this sequence of transitions always exists because when we do not regard the direction of each edge, the given DAG can be considered an undirected connected graph.

**Definition 3.1 (Directed distance in the GFlowNet)** *Let $\tau = (s = s_0 \to s_1 \to \ldots \to s_n = s')$ be the sequence of transitions from $s$ to $s'$ where $s_t \to s_{t+1}$ can be a forward or backward transition. The length of the trajectory $\tau$ is defined as follows:*

$$Len(\tau) := -\log(P(\tau \mid s)), \tag{3}$$

*and the directed distance from $s$ to $s'$ is also defined as follows:*

$$d(s, s') := \min_{\tau = (s \to \ldots \to s')} -\log(P(\tau \mid s)). \tag{4}$$

*where $\tau = (s = s_0 \to s_1 \to \ldots \to s_n = s')$ is a sequence of transitions from $s$ to $s'$ where $s_t \to s_{t+1}$ can be a forward or backward transition and $d(s, s') = 0$ when $s \equiv s'$. Intuitively, $d(s, s')$ is the shortest path length from $s$ to $s'$.*

Although, $d(s, s')$ does not satisfy conditions of being a "distance", it is still a pseudoquasimetric, which motivates us to use the term "directed distance" as an analogy with "directed distance in digraphs" Chartrand & Tian (1997). Indeed, the proposed directed distance is the natural choice for distance in the GFlowNet. Firstly, let's consider two states $s$ and $s'$. They can be considered the equivalent of one another, and have zero distance, if $P(s'|s)$, the probability of transitioning from state $s$ to state $s'$, is equal to $1$. In contrast, if the transition probability is equal to $0$, we cannot reach $s'$ from $s$ by following the GFlowNet policy, then the distance must be infinite. Another reason is that distances are additive, whereas probabilities are multiplicative. Consequently, if we want the length of a trajectory to be related to its likelihood, the transition probabilities between states should be changed to their distance by a "negative" logarithmic scale.

**Why do we use the proposed directed distance as transportation cost to compute OT distance?** Several reasons exist for using the proposed directed distance as transportation cost to compute the OT distance. First, as discussed above, it is the natural choice for turning a transition probability into a distance. Second, with the transportation cost defined as our directed distance, we can decompose the OT distance into entropy and other terms. Consequently, this property not only gives us an upper bound for the OT distance but also imposes sparsity prior on the GFlowNet's structure when minimizing the OT distance or improves exploration when maximizing the OT distance. Besides, the construction of the transportation cost makes minimizing OT distance correspond to maximizing the transition probability between children of neighbor states. Moreover, it also allows us to derive closed-form solutions for the OT distance under some conditions.

**Optimal transport formulation of the path regularization** Once we have defined the directed distance between two states in the GFlowNet, we can define the OT distance between two forward policies of neighbor states. Consider two neighbor states $s$ and $s'$ in trajectory $\tau$, such that $s \to s' \in \mathcal{A}$. The forward policy $P_F(\cdot|s)$ is a discrete probability measure supported by $\text{Child}(s) = \{u_1, ..., u_k\}$ and $P_F(\cdot|s')$ is a discrete probability measure supported by $\text{Child}(s') = \{v_1, ..., v_l\}$.

The OT distance between $P_F(\cdot|s)$ and $P_F(\cdot|s')$ can be defined as:

$$\text{OT}_{\mathbf{C}}\left(P_F(\cdot|s), P_F(\cdot|s')\right) := \min_{\pi \in \prod(P_F(\cdot|s), P_F(\cdot|s'))} \langle \mathbf{C}, \pi \rangle, \tag{5}$$

where the set of admissible couplings is defined as:

$$\Pi(P_F(\cdot|s), P_F(\cdot|s')) := \left\{ \pi \in \mathbb{R}_+^{k \times l} : \pi \mathbb{1}_l = P_F(\cdot|s), \pi^\top \mathbb{1}_k = P_F(\cdot|s') \right\}, \tag{6}$$

and $\mathbf{C}$ is a cost matrix whose each entry is the length of shortest path from $u_i$ to $v_j$:

$$\mathbf{C}_{ij} = c(u_i, v_j) := d(u_i, v_j) = \min_{\tau = (u_i \to ... \to v_j)} -\log(P(\tau \mid u_i)), \tag{7}$$

However, the underlying DAG is unavailable during training progress because of the enormous number of states and edges connecting them. Then $\mathbf{C}_{ij}$ in Eqn. 7 can only be approximately computed by using trajectories in the sub-graph containing $s$, $s'$, their child states, and the edges that connecting them. Specifically, rather than going directly from $u_i$ to $v_j$ if this edge exists, we can always move from $u_i$ to $v_j$ along a back-and-forth trajectory, i.e., $u_i \to s \to s' \to v_j$, with the probability $P_B(s|u_i)P_F(s'|s)P_F(v_j|s')$. Therefore, each entry of the cost matrix $\mathbf{C}$ can be calculated in practice by approximating as follows (note that we abuse the notation of "=" in Eqn. 8 instead of "≈" for easier viewing):

$$\mathbf{C}_{ij} = \begin{cases} 0, & \text{if} \quad u_i \equiv v_j \\ \min\left(-\log(P_B(s \mid u_i)P_F(s' \mid s)P_F(v_j \mid s')), -\log(P(v_j \mid u_i))\right), & \text{else if} \quad u_i \to v_j \in \mathcal{A} \\ -\log(P_B(s \mid u_i)P_F(s' \mid s)P_F(v_j \mid s')), & \text{otherwise.} \end{cases} \tag{8}$$

**Definition 3.2 (Optimal Transport Formulation of the Path Regularization)** *For any complete trajectory $\tau = (s_0 \to s_1 \to ... \to s_n)$, we define the path regularization via OT as follows:*

$$\mathcal{L}_{OT}(\tau) := \sum_{t=0}^{n-1} OT_{\mathbf{C}_{t;\theta}}\left(P_F(\cdot|s_t; \theta), P_F(\cdot|s_{t+1}; \theta)\right). \tag{9}$$

*where $\mathbf{C}_{t;\theta}$ is the cost matrix where each entry is defined in 8.*

If $\pi_\theta$ is the training policy – usually that given by $P_F(\cdot|\cdot; \theta)$ or a modified version of it – then the trajectory loss is updated along trajectories sampled from $\pi_\theta$, i.e., with stochastic gradient:

$$\mathbb{E}_{\tau \sim \pi_\theta} \nabla_\theta (\mathcal{L}_{\text{TB}}(\tau) + \lambda \mathcal{L}_{\text{OT}}(\tau)). \tag{10}$$

where $\lambda \in \mathbb{R}$, $\lambda > 0$ indicates that we want to minimize the path regularization and vice versa.

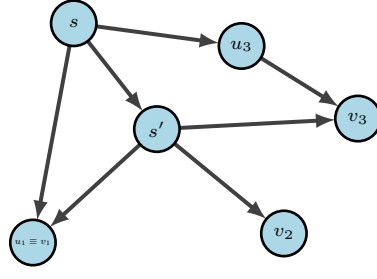

Figure 1: OT distance between $P_F(\cdot|s)$ and $P_F(\cdot|s')$. The forward policy $P_F(\cdot|s)$ is a discrete probability measure supported by Child$(s) = \{u_1, u_2 := s', u_3\}$. Similarly, $P_F(\cdot|s')$ is a discrete probability measure supported by Child$(s') = \{v_1 \equiv u_1, v_2, v_3\}$. The cost matrix is a $3\times3$ matrix. For example, $c_{11} = d(u_1, v_1) = 0$ (because $u_1 \equiv v_1$). There exist many possible paths to move from $u_3$ to $v_3$. First, going directly from $u_3$ to $v_3$ with a distance len$(u_3 \to v_3) = -\log(P_F(v_3|u_3))$. Second, we can move from $u_3$ to $v_3$ along a back-and-forth trajectory, i.e., $u_3 \to s \to s' \to v_3$, with a distance $-\log(P_B(s|u_3)) - \log(P_F(s'|s)P_F) - \log(P(v_3|s'))$. Because the transportation cost from $u_3$ to $v_3$ is the length of the shortest path from $u_3$ to $v_3$, $c_{33} = d(u_3, v_3) \approx \min(-\log(P_B(s|u_3)P_F(s'|s)P_F(v_3|s')), -\log(P_F(v_3|u_3)))$.

**The effect of minimizing the path regularization:** Following Eqn. 5, minimizing $\text{OT}_{\mathbf{C}}(P_F(\cdot|s), P_F(\cdot|s'))$ makes $P_F(v|s')$ closed to $P_F(u|s)$, where $c(u, v)$ small. This is because all probability mass from $u$ can be transferred to $v$ with a smaller cost than other places, reducing the transportation costs' expectation. Also, $P_F(v|s')$ and $P_F(u|s)$ will increase where $c(u, v)$ is small, inducing the similarity of the forward policies and making the flow focus on specific directions. Note that the cost matrix $\mathbf{C}$ is not a constant and depends on the forward policies. Therefore, minimizing the OT distance affects not only the forward policies but also the transportation cost. However, the effect on the forward policies is also consistent with the effect on the transportation costs. As in Eqn. 14 of Theorem 3.2, increasing $P_F(u|s)$ leads to increase $P_B(s|u)$. Besides, $-\log(P_B(s|u)P_F(s'|s)P_F(v|s'))$ is an upper bound of $c(u, v)$, so increasing $P_F(u|s)$ and $P_F(v|s')$ also makes $c(u, v)$ smaller. Also, the OT distance is equal to zero when all probability mass $P_F(v|s') = P_F(u|s)$ concentrate on $u, v$ where $c(u, v) = 0$. For examples, $P_F(s_t|s_{t-1}) = 1$ and $P_F(s_{t+1}|s_t) = 1$ is a special case. When the GFlowNet visits a high reward object, the prior by design helps the GFlowNet quickly adapt its flow to this high reward terminal state. More discussion about relation with entropy is provided in Theorem 3.1.

**The effect of maximizing the path regularization:** In contrast, maximizing makes the forward policies different, so more diverse actions are chosen, leading to more diverse and novel candidates. As in the Theorem 3.1, the upper bound of the OT distance contains the entropy of forward policy $\mathbf{H}(P_F(\cdot|s))$. Moreover, when the training policy is given by the forward policy $P_F$, the upper bound also contains the entropy of the path $\mathbf{H}(P(\tau))$. Thus, maximizing the OT distance is expected to increase the upper bound, which increases the entropy as well. This means more diversity and exploration. Recall that in terms of probabilistic interpretation, we can rewrite the OT distance as the minimum expectation of the transportation costs $\min_{\gamma \sim \Pi(\alpha, \beta)} \mathbb{E}_{u,v \sim \gamma} c(u, v)$. Thus, maximizing the OT distance means maximizing the cost $c(u, v)$. Because $c(u, v)$ is an inverse function of transition probability from $u$ to $v$, maximizing the OT distance means minimizing $P(u \to v)$. Consequently, the flow is distributed to more states, so more diverse actions are chosen.

### 3.2 Upper Bound and Efficient Implementation of the Path Regularization

The cost of computing the OT distance OT$(P_F(\cdot|s), P_F(\cdot|s'))$ scales at least in $\mathcal{O}(d^3 log(d))$, where $d$ is the number of support points. By using the Sinkhorn algorithm (Cuturi (2013)), which solves OT with entropic regularization, we can reduce the computational complexity to $\mathcal{O}(d^2)$ (Altschuler et al., 2017; Lin et al., 2019; 2022). However, our path regularization's definition requires computing the OT distances for all edges in the trajectory $\tau$, which imposes a heavy burden on the computing resources and capacity. To overcome this problem, in Theorem 3.1, we propose the upper bound of the OT distance OT$(P_F(\cdot|s), P_F(\cdot|s'))$. The upper bound provides an efficient implementation and can explain the path regularization's behaviors. The reason is that we can decompose the OT distance into entropy and other terms. Moreover, when the GFlowNet's settings satisfy certain conditions (Section 3.3), we can solve the OT problem with a closed-form solution. Our upper bound and closed-form formulation both have the computational complexity of $\mathcal{O}(LV)$ where $L$ is the maximal length of constructed sequences and $V$ is the action space size.

**Theorem 3.1 (Upper bound of optimal transport distance)** *For any trajectory $\tau = (s_0 \to s_1 \to ... \to s_n)$. The path regularization via OT $\mathcal{L}_{OT}(\tau)$ can be upper bound by:*

$$\mathcal{L}_{UB}(\tau) := \sum_{t=0}^{n-1} \left[ \sum_{u \in Child(s_t)} P_F(u|s_t) \log(P_B(s_t|u)) - \log(P_F(s_{t+1}|s_t)) + \mathbf{H}(P_F(\cdot|s_{t+1})) \right].$$

$$(11)$$

The proof of Theorem 3.1 is provided in Appendix D.1. The entropy terms in the upper bound $\mathcal{L}_{\text{UB}}(\tau)$ encourage the sparsity of the GFlowNet's flow. In addition, when we minimizes the upper bound $\mathcal{L}_{\text{UB}}(\tau)$, the terms $\sum_{u \in \text{Child}(s_t)} P_F(u|s_t) \log(P_B(s_t|u))$ can make $P_B(s_t|u)$ become higher when $P_F(u|s_t)$ is high. As a result, increasing $P_B(s_t|u)$ makes other flows leading to $u$ pruned. While the meanings of the terms $\sum_{u \in \text{Child}(s_t)} P_F(u|s_t) \log(P_B(s_t|u))$ and entropy terms are clear, we now would like to explain the meaning of regularization terms $-\log(P_F(s_{t+1}|s_t))$. Direct calculation indicates that:

$$\sum_{t=0}^{n-1} -\log(P_F(s_{t+1}|s_t)) = -\log\left(\prod_{t=0}^{n-1} P_F(s_{t+1}|s_t)\right) = -\log(P(\tau)), \qquad (12)$$

Moreover, when the training policy $\pi_\theta$ is given by the forward policies $P_F(.|.;\theta)$, we have:

$$\mathbb{E}_{\tau \sim \pi_\theta}(-\log(P(\tau))) = \mathbf{H}(P(\tau)). \qquad (13)$$

By taking this approach, the upper bound regularizes not only on the forward policy via $\mathbf{H}(P_F(\cdot|s_{t+1}))$ but also on the path via $\mathbf{H}(P(\tau))$. Besides, we can think $-\log(P(\tau)) = -\log(P(\tau|s_0)P(s_0)) = -\log(P(\tau|s_0))$ as the length of $\tau$. Then minimizing path regularization can result in shorter paths and smaller numbers of paths with high flows.

### 3.3 CLOSED-FORM FORMULATION FOR THE PATH REGULARIZATION

In some specific circumstances, i.e., synthetic hypergrid environment (Bengio et al. (2021a)), discrete probabilistic modeling (Zhang et al. (2022)), and biological sequence design (Jain et al. (2022)), Theorem 3.2 allows us to compute OT loss by a closed-form formulation, where the cost matrix $\mathbf{C}$ is re-defined by its approximation as in Eqn. 8. Specifically, the closed-form formulation was derived by taking advantage of the following observations. First, in these cases, each two neighbor states $s$ and $s'$ on each sampled trajectory, such that $s \to s' \in \mathcal{A}$, don't share any child state. In other words, there doesn't exist any state $s''$ satisfying $s \to s'' \in \mathcal{A}$ and $s' \to s'' \in \mathcal{A}$. This finding is resulted from the property that each action can not decompose into the composition of others in the action space. Second, if there exists an edge connecting from a child state $u$ of $s$, such that $u \neq s'$, to a child state $v$ of $s'$ then the action of transition from $s$ to $u$ must be the same action of of transition from $s'$ to $v$ and the transition action from $u$ to $v$ must be exactly the transition action from $s$ to $s'$.

**Theorem 3.2 (Closed-form solution for optimal transport distance)** *Let the OT cost between the forward policies of two neighbor states be defined as in Eqn. 5 where the cost matrix $\mathbf{C}$ is re-defined by its approximation as in Eqn. 8. For each non-terminal neighbor states $s$ and $s'$ such that $s \to s' \in \mathcal{A}$, let $a_i$ be an action so that the state-action pair $(s, a_i)$ leads to $u_i$ and $(s', a_i)$ leads to $v_i$, where $u_i \in Child(s)$ and $v_i \in Child(s')$. Let $A_s^*$, $A_{s'}^*$ be the set of non-terminal valid actions at state $s$, $s'$ and $a_s^*$ be the action of moving from $s$ to $s'$. If the following conditions are satisfied: (1) $a_i \neq a_k + a_h \quad \forall a_i, a_k, a_h \in \mathcal{A}$, and (2) if $\exists \quad a_i, a_h, a_m, a_n \in \mathcal{A}$ such that $a_i + a_h = a_m + a_n, a_i \neq a_m$ then $a_i = a_n, a_h = a_m$; the following result holds:*

$$OT(P_F(\cdot|s), P_F(\cdot|s')) = \sum_{u \in Child(s)} P_F(u|s) \log(P_B(s|u)) + \mathbf{H}(P_F(\cdot|s'))$$
$$+ P_F(s'|s)(\log(P_B(s'|s)) + \log(P_F(s'|s))) \qquad (14)$$
$$+ \sum_{a_i \in A_s^* \bigcap A_{s'}^*, u_i \neq s', u_i + a_s^* = v_i} \min(P_F(u_i|s), P_F(v_i|s'))c_i',$$

*where we define:*

$$c_i' = \begin{cases} \min(0, \log(P_B(s|u_i)) + \log(P_F(s'|s)) + \log(P_F(v_i|s')) - \log(P_F(v_i|u_i))), & \text{if } u_i \neq s' \\ 0, & \text{if } u_i = s' \end{cases} \qquad (15)$$

The proof of Theorem 3.2 and the closed-form solution for OT distance at terminal states are provided in Appendix D.2. These closed-form solutions of the OT distance will be used in our experiments in Section 4 (see Appendix D.2 for the reasons).

## 4 EXPERIMENTAL RESULTS

In this section, we numerically justify the advantage of OT regularization over the baseline GFlowNet model only trained with trajectory balance loss on a wide range of tasks: hypergrid environment, discrete probabilistic modeling, and biological sequence design tasks. We aim to show

that: (i) Minimizing the path regularization via OT improves the GFlowNets' generalization, and the upper bound can be used as an efficient approximation when we want to minimize the regularization term (hyper-grid environment, discrete probabilistic modeling); while (ii) Maximizing path regularization via OT enhances the exploration ability of the GFlowNet (biological sequence tasks).

## 4.1 HYPER-GRID ENVIRONMENT

**Task** We follow the framework of Malkin et al. (2022) with slight changes to study a hyper-grid environment, which evaluates the generalization ability of the GFlowNet to guess and sample unvisited modes of the interested distribution. Consider a $D$-dimensional hyper-grid environment with length of each side is $H$, where each cell represents non-terminal state of the given DAG: $s = (s_1, \ldots, s_D)$ where $s_d \in \{0, 1, \ldots, H-1\}$ for $d \in \{1, \ldots, D\}$. The source state is $(0, 0, ..., 0)$. For any non-terminal state, the available actions are operations of increasing coordinate $i$ by 1 that still satisfies $s_i \leq H - 1$ and a terminating action that moves to a corresponding terminal state $s^T$, which has its reward:

$$R\left(s^\top\right) = R_0 + 0.5 \prod_{d=1}^{D} \mathbb{I}\left[|s_d/(H-1) - 0.5| \in (0.25, 0.5]\right] + 2 \prod_{d=1}^{D} \mathbb{I}\left[|s_d/(H-1) - 0.5| \in (0.3, 0.4)\right]$$

(16)

where $R_0$ is the constant that controls the discovery challenge and $\mathbb{I}$ is the indicator function. This reward function indicates that only considerable rewards exist at the environment's corners, and there are correct $2^D$ modes. The experiment is conducted for two hyper-grid environments with the number of dimensions 4 and 8 (a higher number of dimensions means more challenging). We consider the same side length $H = 8$ and $R_0 = 10^{-3}$ for both environments. To evaluate the performance on this task, we use KL divergence and the number of modes found during training as the main evaluation metrics. More details about architectures, hyper-parameters, and evaluation criteria are provided in Appendix E.1.

**Results** We plot the mean results over 10 runs for each configuration in Fig. 2. Generally, the observed behaviors of the proposed regularization methods become more apparent when the hypergrid environment is sparser. Precisely, although recovering full modes, the GFlowNet model trained by minimizing the path regularization via OT discovers modes faster than the baseline, which indicates its focus on finding directions leading to states with high rewards during the training progress rather than spending time exploring the environment. This also helps the model better discover the latent structures of the interested distribution and achieve lower KL error. We can also see that the upper bound is an efficient approximation in terms of complexity when using a positive regularization coefficient, whose performance is even better. Meanwhile, Max OT seems unsuitable because of its motivation to improve the model's exploration, while we only need the high-reward states near the corners, and the majority of states have minimal rewards.

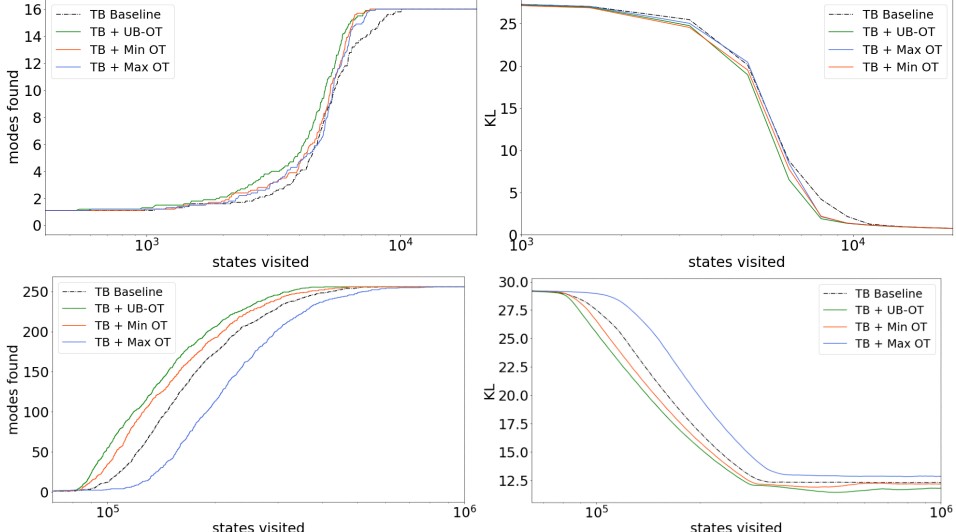

Figure 2: Results on the $4 - D$ (upper) and $8 - D$ (lower) hyper-grid environment. Left: Number of modes found during training. Right: KL divergence between the true and empirical distribution.

## 4.2 BIOLOGICAL SEQUENCE DESIGN

**Task** We follow the framework of Jain et al. (2022) to simulate the process of designing biological sequences, such as anti-microbial peptides (AMP), DNA and protein sequences (TF Bind 8, GFP). The experiments are conducted in the multi-round active learning setting, with the goal of generating a diverse set of useful candidates after evaluation rounds. We report the performance score, diversity score, and novelty score of the TopK scoring candidates to evaluate the performance of each method. More details about task description, datasets, hyper-parameters, and evaluation criteria are provided in Appendix E.2.

|  | Performance | Diversity | Novelty |
|---|---|---|---|
| DynaPPO | $\mathbf{0.938 \pm 0.009}$ | $12.12 \pm 1.71$ | $9.31 \pm 0.69$ |
| COMs | $0.761 \pm 0.009$ | $19.38 \pm 0.14$ | $26.47 \pm 1.30$ |
| GFlowNet-AL (paper) | $0.932 \pm 0.002$ | $22.34 \pm 1.24$ | $28.44 \pm 1.32$ |
| GFlowNet-AL | $0.874 \pm 0.022$ | $\mathbf{31.98 \pm 2.27}$ | $23.91 \pm 1.87$ |
| GFlowNet+Min OT-AL | $0.847 \pm 0.033$ | $20.32 \pm 7.38$ | $23.63 \pm 1.66$ |
| GFlowNet+UB OT-AL | $0.828 \pm 0.022$ | $29.89 \pm 2.80$ | $24.16 \pm 1.75$ |
| GFlowNet+Max OT-AL | $0.917 \pm 0.003$ | $\mathbf{31.56 \pm 2.43}$ | $\mathbf{28.86 \pm 0.96}$ |

Table 1: Results on the AMP task with $K = 100$.

**AMP** The results for AMP design task is shown in Table 1. We see that the GFlowNet-AL model trained by maximizing the regularization via OT performs better than other baselines in terms of diversity and novelty. In addition, the TopK performance of the GFLowNet-AL baseline also increases from $0.874$ to $0.917$ when we maximize the OT regularization and is only lower than the reported performance of DynaPPO. However, DynaPPO has a much lower diversity and novelty score, which implies that it mostly generates similar candidates from the training dataset.

**TF Bind 8** An interesting observation here is that initial dataset $\mathcal{D}_0$ contains only half of all possible DNA sequences of length $8$ having lower scores. Specifically, low-quality data is very common in practice, and in this task, it poses a big challenge for all the methods to have good results. From Table 2, we can see that MINs have the highest diversity compared to the other methods. However, this method has a much lower TopK performance and novelty score, which indicates its generated samples are very similar to the low-quality training dataset. Moreover, although having slightly lower diversity, the GFlowNet-AL model trained by maximizing the path regularization via OT performs better than the others when looking at all metrics - it outperforms other baselines in terms of performance and novelty score.

|  | Performance | Diversity | Novelty |
|---|---|---|---|
| DynaPPO | $0.58 \pm 0.02$ | $5.18 \pm 0.04$ | $0.83 \pm 0.03$ |
| COMs | $0.74 \pm 0.04$ | $4.36 \pm 0.24$ | $1.16 \pm 0.11$ |
| BO-qEI | $0.44 \pm 0.05$ | $4.78 \pm 0.17$ | $0.62 \pm 0.23$ |
| CbAS | $0.45 \pm 0.14$ | $5.35 \pm 0.16$ | $0.46 \pm 0.04$ |
| MINs | $0.40 \pm 0.14$ | $\mathbf{5.57 \pm 0.15}$ | $0.36 \pm 0.00$ |
| CMA-ES | $0.47 \pm 0.12$ | $4.89 \pm 0.01$ | $0.64 \pm 0.21$ |
| AmortizedBO | $0.62 \pm 0.01$ | $4.97 \pm 0.06$ | $1.00 \pm 0.57$ |
| GFlowNet-AL (paper) | $0.84 \pm 0.05$ | $4.53 \pm 0.46$ | $2.12 \pm 0.04$ |
| GFlowNet-AL | $0.83 \pm 0.01$ | $4.66 \pm 0.08$ | $1.14 \pm 0.03$ |
| GFlowNet+Min OT-AL | $0.82 \pm 0.01$ | $4.72 \pm 0.10$ | $1.13 \pm 0.04$ |
| GFlowNet+UB OT-AL | $0.83 \pm 0.01$ | $4.68 \pm 0.10$ | $1.14 \pm 0.05$ |
| GFlowNet+Max OT-AL | $\mathbf{0.85 \pm 0.02}$ | $4.52 \pm 0.18$ | $\mathbf{1.21 \pm 0.10}$ |

Table 2: Results on the TF Bind 8 task with $K = 128$.

**GFP** Lastly, the results for the GFP task are shown in Table 3, where the objective is to find protein sequences having high fluorescence. We observe that the GFlowNet-AL model trained by maximizing OT regularization generates more diverse and novel candidates than other methods. In addition, its performance score is only lower than the best one achieved by COMs and higher than the GFlowNet-AL baseline. However, when looking at all metrics, the GFlowNet-AL model trained by maximizing the path regularization via OT still outperforms all other baselines.

Note that in all biological sequence tasks, Min OT and UB OT do not improve the performance of GFlowNets since these methods make the forward policy at each state have a focused tendency, which seems not to increase the diversity and novelty of the generated candidates.

## 4.3 SYNTHETIC DISCRETE PROBABILISTIC MODELING TASKS

**Task** We follow the framework of Zhang et al. (2022), Energy-based Generative Flow Networks (EB-GFN), to model seven different distributions over 32-dimensional binary data that are dis-

| | Performance | Diversity | Novelty |
|---|---|---|---|
| DynaPPO | $0.794 \pm 0.002$ | $206.19 \pm 0.19$ | $203.20 \pm 0.47$ |
| COMs | $\mathbf{0.831 \pm 0.003}$ | $204.14 \pm 0.14$ | $201.64 \pm 0.42$ |
| BO-qEI | $0.045 \pm 0.003$ | $139.89 \pm 0.18$ | $203.60 \pm 0.06$ |
| CbAS | $0.817 \pm 0.012$ | $5.42 \pm 0.18$ | $1.81 \pm 0.16$ |
| MINs | $0.761 \pm 0.007$ | $5.39 \pm 0.00$ | $2.42 \pm 0.00$ |
| CMA-ES | $0.063 \pm 0.003$ | $201.43 \pm 0.12$ | $203.82 \pm 0.09$ |
| AmortizedBO | $0.051 \pm 0.001$ | $205.32 \pm 0.12$ | $202.34 \pm 0.25$ |
| GFlowNet-AL (paper) | $0.853 \pm 0.004$ | $211.51 \pm 0.73$ | $210.56 \pm 0.82$ |
| GFlowNet-AL | $0.8232 \pm 0.0001$ | $218.54 \pm 7.88$ | $222.05 \pm 5.49$ |
| GFlowNet+Min OT-AL | $0.8231 \pm 0.0001$ | $182.03 \pm 0.25$ | $220.56 \pm 2.02$ |
| GFlowNet+UB OT-AL | $0.8232 \pm 0.0001$ | $221.64 \pm 0.11$ | $218.02 \pm 0.79$ |
| GFlowNet+Max OT-AL | $0.8233 \pm 0.0001$ | $\mathbf{225.00 \pm 3.76}$ | $\mathbf{242.11 \pm 1.44}$ |

Table 3: Results on the GFP task with $K = 128$.

cretizations of continuous distributions over the plane. The state space $\mathcal{S}$ of the GFlowNet consists of vectors of length $D = 32$ with with entries in $\{0; 1; \oslash\}$. The source state is $s_0 = (\oslash, \oslash, ..., \oslash)$. For any non-terminal state, the available actions are turning a void entry $\oslash$ to 0 or 1. After $D$ actions, we reach the terminal states having all entries in $\{0; 1\}$. The main evaluation metrics are NLL score and MMD score. The detailed settings about architectures, hyper-parameters, and evaluation criteria are provided in Appendix E.3.

**Results** The results for synthetic discrete probabilistic modeling tasks (Synthetic EB-GFN) are shown in Table 4. Training the GFlowNet with either minimizing the path regularization via OT (Min OT) or via the upper bound (UB OT) gains the better NLL and MMD scores than the baseline and Max OT. We also observe that the performance of training EB-GFN with Min OT and UB OT are quite similar. Meanwhile, Max OT is not useful due to the same reasons provided in hypergrid environment modeling task. Because there is a gap between our reproduce results and the baseline in EB-GFN Zhang et al. (2022), we only take into account the the reproduce results of EB-GFN when comparing with our methods (Min OT, Max OT, and UB OT).

| Metric | Method | 2spirals | 8gaussians | circles | moons | pinwheel | swissroll | checkerboard |
|---|---|---|---|---|---|---|---|---|
| NLL ↓ | PCD | 20.094 | 19.991 | 20.565 | 19.763 | 19.593 | 20.172 | 21.214 |
| | ALOE | 20.295 | 20.350 | 20.565 | 19.287 | 19.821 | 20.160 | 54.653 |
| | ALOE + | 20.062 | 19.984 | 20.570 | 19.743 | 19.576 | 20.170 | 21.142 |
| | EB-GFN (paper) | 20.050 | 19.982 | 20.546 | 19.732 | 19.554 | 20.146 | 20.696 |
| | EB-GFN | 20.0679 | 19.9862 | **20.5598** | 19.7324 | 19.5735 | 20.1599 | 20.6839 |
| | EB-GFN + Max OT | 20.0673 | 19.9857 | 20.5599 | 19.7319 | 19.5714 | 20.1597 | 20.6837 |
| | EB-GFN + UB OT | 20.0651 | **19.9854** | 20.5600 | **19.7305** | 19.5707 | 20.1596 | 20.6836 |
| | EB-GFN + Min OT | **20.0640** | 19.9855 | **20.5598** | 19.7308 | **19.5699** | **20.1595** | **20.6831** |
| MMD ↓ | PCD | 2.160 | 0.954 | 0.188 | 0.962 | 0.505 | 1.382 | 2.831 |
| | ALOE | 21.926 | 107.320 | 0.497 | 26.894 | 39.091 | 0.471 | 61.562 |
| | ALOE + | 0.149 | 0.078 | 0.636 | 0.516 | 1.746 | 0.718 | 12.138 |
| | EB-GFN (paper) | 0.583 | 0.531 | 0.305 | 0.121 | 0.492 | 0.274 | 1.206 |
| | EB-GFN | 0.3012 | 0.0408 | $-0.1724$ | $-0.1744$ | 0.2056 | 0.1555 | $-0.0986$ |
| | EB-GFN + Max OT | 0.3258 | 0.0197 | $-0.1919$ | $-0.0456$ | 0.1377 | 0.0763 | $-0.0903$ |
| | EB-GFN + UB OT | 0.2902 | **0.0102** | **$-0.2819$** | $-0.1253$ | 0.1561 | **0.0257** | $-0.0923$ |
| | EB-GFN + Min OT | **0.1816** | 0.0343 | $-0.2775$ | **$-0.1966$** | **0.1220** | 0.1334 | **$-0.1071$** |

Table 4: Results on the Synthetic EB-GFN tasks. The negative log-likelihood (NLL) and MMD are displayed in units of $1 \times 10^{-4}$. ALOE+ uses a 30 larger parametrization than ALOE and EB-GFN. We only take into account the the reproduce results of EB-GFN when comparing with our methods (Min OT, Max OT, and UB OT).

## 5 CONCLUDING REMARKS

In this paper, we propose to train the GFlowNet with an additional path regularization via Optimal Transport that places prior constraints on the underlying structure of the GFlowNet. We have empirically shown that minimizing the path regularization via OT improves the GFlowNet's generalization while maximizing path regularization via OT enhances the exploration ability of the GFlowNet. In addition, we derive an efficient implementation of the regularization by finding its closed-form solutions in specific cases and a meaningful upper bound that can be used as an approximation when we want to minimize the regularization term. A limitation of the current method is computing the optimal transport distances for all couples of nearest neighbor states. Our proposed Dropout OT (see in Appendix C) might be a solution. In future works, we aim to develop a more efficient path regularization for high dimensional discrete data or propose a new cost function to compute the optimal transport distances.

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

# Supplement to "Improving Generative Flow Networks with Path Regularization"

## A  RELATED WORK

**GFlowNets** The objective of GFlowNets is related to MCMC methods for sampling from a given unnormalized density function, especially in discrete spaces where exact sampling is intractable (Dai et al. (2020); Grathwohl et al. (2021)). However, GFlowNets amortize the complexity of iterative sampling by a training procedure that implies the data's compositional structure as its learning problem. Empirically, GFlowNets' performance is better than other earlier methods in a wide variety of tasks: small molecules generation (Bengio et al. (2021a)), discrete probabilistic modeling (Zhang et al. (2022)), Bayesian structure learning (Deleu et al. (2022)) and biological sequence design (Jain et al. (2022)). On the theoretical side, definitions and properties of GFlowNets are more investigated in Bengio et al. (2021b).

**Optimal Transport** The optimal transport theory (OT) (Villani (2003)) has established a natural and useful geometric tool for comparing measures supported on metric probability spaces. The development of OT theory has a long history, where it has been discovered in many settings and under different forms. And in recent years, another revolution in the spread of OT has been witnessed, thanks to the emergence of approximate solvers that can scale to the problem of large dimensions. As a consequence, OT is being widely used to solve various problems in computer graphics (Bonneel et al. (2011),Nguyen et al. (2021)), image processing (Xia et al. (2014)), and machine learning (Courty et al. (2014), Ho et al. (2017) Genevay et al. (2018), Bunne et al. (2019)).

**Energy-based models** EBMs, or energy functions parameterized by deep neural networks, have demonstrated effectiveness in generative modeling (Salakhutdinov & Hinton (2009); Hinton et al. (2006)). Contrastive divergence methods (Hinton (2002); Tieleman (2008); Du et al. (2021)) have been proposed to handle costly MCMC processes by approximating energy gradient. Recently, it has been shown that simultaneous learning of the proposal distribution can also be helpful (Dai et al. (2019); Arbel et al. (2021)). Then this finding has been extended to discrete spaces by using GFlowNets in Zhang et al. (2022).

**Biological sequence design** Various methods have been proposed to handle the biological sequence design tasks: deep model-based optimization (Trabucco et al. (2021)), Bayesian optimization (Belanger et al. (2019); Swersky et al. (2020)), reinforcement learning (Angermueller et al. (2020)), adaptive evolutionary methods (Hansen (2006); Sinai et al. (2020)), and so on. Recently, GFlowNets also have been proposed as a useful generator of diverse candidates for this problem in Jain et al. (2022).

## B  BACKGROUND OF GFLOWNETS

Generative Flow Networks (GFlowNets) are a recently proposed class of generative model, which aims to sample a structural object $\mathbf{x}$ with probability proportional to a given reward function $R(\mathbf{x})$. From the reinforcement learning viewpoint, GFlowNets learn a stochastic policy to generate object $\mathbf{x} \in \mathcal{X}$ by applying a sequence of discrete actions $a \in \mathcal{A}$ where $\mathcal{A}$ is the action space. The construction of an object $x \in \mathcal{X}$ defines a *complete trajectory* $\tau = (s_0, s_1, ..., s_n = x, s_f)$ where $s_0$ is the *initial state*, $s_n = x \in \mathcal{X}$ is the *terminal state* (indicating entirely constructed object), and $s_f$ is the *final state*. Note that the same terminal state can be formed by different sequences of actions. These states and actions correspond to the vertices and edges of a directed acyclic graph $G = (\mathcal{S}, \mathcal{A})$. In addition, for each transition $s \to s' \in \mathcal{A}$, we call $s$ a parent of $s'$, and $s'$ a child of $s$. $\mathcal{T}$ is defined as the set of all complete trajectories.

Following Bengio et al. (2021b), a *trajectory flow* is any nonnegative function defined on the set of complete trajectories, such as $F : \mathcal{T} \mapsto \mathbb{R}^+$. Correspondingly, the flow through a state (state flow) is defined as $F(s) = \sum_{\tau \in \mathcal{T}, s \in \tau} F(\tau)$ and the flow through a edge (edge flow) is defined as $F(s \to s') = \sum_{\tau \in \mathcal{T}, s \to s' \in \tau} F(\tau)$. Additionally, the forward transition probabilities $P_F$ and the backward transition probabilities $P_B$ are defined as follows:

$$P_F(s'|s) := \frac{F(s \to s')}{F(s)}, \tag{17}$$

$$P_B(s|s') := \frac{F(s \rightarrow s')}{F(s')}. \tag{18}$$

Then the training objective of the GFlowNet is to learn a *consistent flow* (Bengio et al. (2021b); Malkin et al. (2022)) that has the *terminal flow* $F(x \rightarrow s_f)$ approximately equal a given reward function $R(x)$ for any $x \in X$. In addition, when the flow is consistent, the forward transition probabilities $P_F$ and the backward transition probabilities $P_B$ correspondingly define a distribution over the children and parent of each state, which can be considered as the forward and backward policy of GFLowNets.

Specifically, followed by Malkin et al. (2022), the GflowNet models the forward policy, backward policy and total flow of a Markovian flow $F$ by $P_F(.|.;\theta)$, $P_B(.|.;\theta)$ and $Z_\theta$. The *trajectory balance objective* is then optimized for each complete trajectory $\tau$ sampled from the training policy $\pi_\theta$:

$$\mathcal{L}_{TB}(\tau, \theta) = \left( \log(Z_\theta \prod_{t=1}^{n} P_F(s_t|s_{t-1}; \theta)) - \log(R(x) \prod_{t=1}^{n} P_B(s_{t-1}|s_t; \theta)) \right)^2. \tag{19}$$

which is derived from the *trajectory balance constraint* (Malkin et al. (2022))

Moreover, as already proved by Bengio et al. (2021b), $\pi_\theta$ can be chosen as any distribution on the set of complete trajectories $\mathcal{T}$ with full supports, or the GflowNet can be trained with offline policy as well, such as a mixture between the GFlowNet's forward policy and an uniform distribution over allowed actions in each state:

$$\pi_\theta = (1 - \alpha) P_F(.|.;\theta) + \alpha \text{ Uniform} \tag{20}$$

There also exist other objectives for learning a GFlowNet, which are based on *flow matching* constraint or *detail balance* constraint as in Bengio et al. (2021a;b). However, Malkin et al. (2022) empirically shows that the trajectory balance objective improves the training of a GFlowNet in terms of more efficient credit assignment and faster convergence, compared to the previously proposed objectives. These advantages make us choose it as the training objective in this paper.

## C  DROPOUT OPTIMAL TRANSPORT

A limitation of the current method is computing the optimal transport distances for all couples of nearest neighbor states, especially in high dimensional discrete data. Our proposed dropout OT might be a solution. This is because rather than sampling trajectories $\tau$ and using all edges from them, we can separately sample edges $s \rightarrow s'$ proportional to edge flows, allowing us to efficiently compute path regularization.

**Theorem C.1** *For any complete trajectory $\tau = (s_0 \rightarrow s_1 \rightarrow ... \rightarrow s_n)$ sampled from the training policy $\pi_\theta$*

$$\mathbb{E}_{\tau \sim \pi_\theta}(\mathcal{L}_{OT}(\tau)) \propto \mathbb{E}_{s \rightarrow s' \sim \pi_\theta}(OT(P_F(\cdot|s), P_F(\cdot|s'))). \tag{21}$$

The proof of Theorem C.1 is in Appendix D.3. Here we train GFlowNets with trajectory balance objective. Therefore, when sampling a trajectory $\tau$, we get a set of edges from $\tau$. We just sample uniformly a $p$ percentage of edges to compute OT loss.

To sample $p$ percentage of edges, let sample $r_s \sim Ber(p)$.

$$\mathbb{E}_{s \rightarrow s' \sim \pi_\theta}(\text{OT}(P_F(\cdot|s), P_F(\cdot|s'))) = \frac{1}{p} \mathbb{E}_{r_s \sim Ber(p)} \mathbb{E}_{s \rightarrow s' \sim \pi_\theta}(r_s . \text{OT}(P_F(\cdot|s), P_F(\cdot|s'))). \tag{22}$$

We approximate the path regularization loss via:

$$\mathcal{L}_{\text{OT}}(\tau) \simeq \frac{1}{p} \sum_{t=0}^{n-1} x_t \text{OT}(P_F(.|s_t), P_F(.|s_{t+1})) \tag{23}$$

with $x_t$ drawn independently from $\text{Ber}(p)$ for all $0 \leq t \leq n - 1$. Intuitively, if $x_t = 0$ then we don't need to calculate the corresponding optimal transport cost anymore, which reduces a considerable amount of computing time and memory down to $p$ percentage.

## D  PROOFS

### D.1  PROOF OF THEOREM 3.1

For any trajectory $\tau = (s_0 \rightarrow s_1 \rightarrow ... \rightarrow s_n)$, we first prove that for any $t \in \overline{0, n - 1}$

$$\text{OT}(P_F(\cdot|s_t), P_F(\cdot|s_{t+1})) \leq \sum_{u \in \text{Child}(s_t)} P_F(u|s_t) \log(P_B(s_t|u)) - \log(P_F(s_{t+1}|s_t)) + \mathbf{H}(P_F(\cdot|s_{t+1})). \tag{24}$$

Consider two neigboor states $s_t$ and $s_{t+1}$ with the children sets: $\text{Child}(s_t) = \{u_1, ..., u_k\}$ and $\text{Child}(s_{t+1}) = \{v_1, ..., v_l\}$. By definition 5, the optimal transportation distance between two distributions $P_F(.|s_t)$ and $P_F(.|s_{t+1})$ is defined as:

$$\text{OT}_{\mathbf{C}}\left(P_F(\cdot|s_t), P_F(\cdot|s_{t+1})\right) := \min_{\pi \in \prod(P_F(\cdot|s_t), P_F(\cdot|s_{t+1}))} \langle \mathbf{C}, \pi \rangle, \tag{25}$$

where the admissible couplings set is defined as:

$$\Pi(P_F(.|s_t), P_F(.|s_{t+1})) = \left\{ \pi \in \mathbb{R}_+^{k \times l} : \pi \mathbb{1}_l = P_F(\cdot|s_t), \pi^{\mathsf{T}} \mathbb{1}_k = P_F(\cdot|s_{t+1}) \right\}. \tag{26}$$

We have,

$$
\begin{aligned}
&\text{OT}(P_F(.|s_t), P_F(.|s_{t+1})) \\
&\leq \sum_i \sum_j \pi_{ij} \mathbf{C}_{ij} \\
&\leq -\sum_i \sum_j \pi_{ij} \log\left(P_B(s_t|u_i) P_F(s_{t+1}|s_t) P_F(v_j|s_{t+1})\right) \\
&= -\sum_i \sum_j \pi_{ij} \log\left(P_B(s_t|u_i)\right) - \sum_i \sum_j \pi_{ij} \log\left(P_F(s_{t+1}|s_t)\right) - \sum_i \sum_j \pi_{ij} \log\left(P_F(v_j|s_{t+1})\right) \\
&= -\sum_i \log\left(P_B(s_t|u_i)\right) \sum_j \pi_{ij} - \log\left(P_F(s_{t+1}|s_t)\right) \sum_i \sum_j \pi_{ij} - \sum_i \log\left(P_F(v_j|s_{t+1})\right) \sum_j \pi_{ij} \\
&= -\sum_i \log\left(P_B(s_t|u_i)\right) P_F(u_i|s_t) - \log\left(P_F(s_{t+1}|s_t)\right) - \sum_j \log\left(P_F(v_j|s_{t+1})\right) P_F(v_j|s_{t+1}) \\
&= \sum_{u \in \text{Child}(s_t)} P_F(u|s_t) \log(P_B(s_t|u)) - \log(P_F(s_{t+1}|s_t)) + \mathbf{H}(P_F(.|s_{t+1}).
\end{aligned}
\tag{27}
$$

The first inequality obtained by the definition of optimal transport distance in Eq. 25, the second inequality comes from Eq. 8, the fifth equality is due to the constraints of admissible couplings in Eq. 26.

As a consequence, the upper bound loss is obtained by summing up all inequality 24 for all $t$.

## D.2 PROOF OF THEOREM 3.2

Recall from definition 5 the optimal transportation distance between two distributions $P_F(.|s)$ and $P_F(.|s')$ is defined as:

$$\text{OT}_{\mathbf{C}}\left(P_F(\cdot|s), P_F(\cdot|s')\right) := \min_{\pi \in \Pi(P_F(\cdot|s), P_F(\cdot|s'))} \langle \mathbf{C}, \pi \rangle. \tag{28}$$

Let decompose the total cost $\langle \mathbf{C}, \pi \rangle$

$$
\begin{aligned}
\langle \mathbf{C}, \pi \rangle &= \sum_{i,j} \pi_{ij} \mathbf{C}_{ij} \\
&= \sum_{i,j} \pi_{ij}(-\log(P_B(s|u_i)) - \log(P_F(s'|s)) - \log(P_F(v_j|s'))) \\
&\quad + \sum_{u_i = s', j} \pi_{ij}(\log(P_B(s|s')) + \log(P_F(s'|s))) \\
&\quad + \sum_{u_i \neq s', v_j \in Child(u_i), a_i \neq a^{\mathsf{T}}} \pi_{ij}(\log(P_B(s|u_i)) + \log(P_F(s'|s)) + \log(P_F(v_j|s') + \mathbf{C}_{ij}) \\
&\quad + \sum_{u_i = v_j} \pi_{ij}(\log(P_B(s|u_i)) + \log(P_F(s'|s)) + \log(P_F(v_j|s'))).
\end{aligned}
\tag{29}
$$

We will prove that $u_i \neq v_j$ $\forall i, j$, i.e, $\text{Child}(s) \cap \text{Child}(s') = \emptyset$, indeed,

$$a_i \neq a_k + a_h \quad \forall a_i, a_k, a_h \in \mathcal{A} \implies a_i \neq a_s^* + a_j \implies s + a_i \neq s + a_s^* + a_j \implies u_i \neq v_j. \quad \forall i, j \tag{30}$$

We have:

$$
\begin{aligned}
& u_i \neq s', \quad v_j \in Child(u_i), \quad a_i \neq a^\top \\
\implies \quad & a_i \neq a_s^*, \quad s + a_i + a_{u_i}^* = s + a_s^* + a_j, \quad a_i \neq a^\top \\
\implies \quad & a_i \neq a_s^*, \quad a_i + a_{u_i}^* = a_s^* + a_j, \quad a_i \neq a^\top \\
\implies \quad & a_i \neq a_s^*, \quad a_i = a_j \neq a^\top, \quad a_{u_i}^* = a_s^*.
\end{aligned}
\tag{31}
$$

As a result, we can rewrite Eq. 29 as:

$$
\begin{aligned}
\langle \mathbf{C}, \pi \rangle = & \sum_{i,j} \pi_{ij}(-\log(P_B(s|u_i)) - \log(P_F(s'|s)) - \log(P_F(v_j|s'))) \\
& + \sum_{u_i = s', j} \pi_{ij}(\log(P_B(s|s')) + \log(P_F(s'|s))) \\
& + \sum_{u_i \neq s', a_i = a_j \neq a^\top} \pi_{ij}(\log(P_B(s|u_i)) + \log(P_F(s'|s)) + \log(P_F(v_j|s') + \mathbf{C}_{ii}).
\end{aligned}
\tag{32}
$$

The first term of above equation actually is the upper bound of the optimal transport distance. Therefore, we can rewrite the total transportation cost as:

$$
\begin{aligned}
\langle \mathbf{C}, \pi \rangle = & \sum_{u \in \text{Child}(s)} P_F(u|s)\log(P_B(s|u)) - \log(P_F(s'|s)) + \mathbf{H}(P_F(\cdot|s')) \\
& + P_F(s'|s).(\log(P_B(s'|s)) + \log(P_F(s'|s))) \\
& + \sum_{u_i \neq s', a_i = a_j \neq a^\top} \pi_{ij}(\log(P_B(s|u_i)) + \log(P_F(s'|s)) + \log(P_F(v_j|s') + \mathbf{C}_{ii}).
\end{aligned}
\tag{33}
$$

From the definition of $c_i'$ in Eq. 15, we have

$$
c_i' = \begin{cases} \log(P_B(s|u_i)) + \log(P_F(s'|s)) + \log(P_F(v_j|s') + \mathbf{C}_{ii}, & \text{if } u_i \neq s', a_i = a_j \neq a^\top \\ 0 & \text{if } u_i = s' \text{ or } a_i = a^\top. \end{cases}
\tag{34}
$$

From Eq. 33 and Eq. 34, we have

$$
\sum_{u_i \neq s', a_i = a_j \neq a^\top} \pi_{ij}(\log(P_B(s|u_i)) + \log(P_F(s'|s)) + \log(P_F(v_j|s') + \mathbf{C}_{ii}) = \sum_i \pi_{ij}.c_i'.
\tag{35}
$$

Thus, we find that

$$
\underset{\pi \in \prod(P_F(\cdot|s), P_F(\cdot|s'))}{\arg\min} \langle \mathbf{C}, \pi \rangle = \underset{\pi \in \prod(P_F(\cdot|s), P_F(\cdot|s'))}{\arg\min} \langle \mathbf{C}', \pi \rangle
\tag{36}
$$

where, $\mathbf{C}'$ is a diagonal matrix with the diagonal $c_i' \leq 0$. For convenience, if action $a_i$ is invalid at state $s$, we assign $P_F(u_i \mid s) := 0$, so the cost matrix of the optimal transport distance still is a square matrix with the zero cost a invalid actions, then applying the Lemma 1, we have:

$$
\underset{\pi \in \prod(P_F(\cdot|s), P_F(\cdot|s'))}{\min} \langle \mathbf{C}', \pi \rangle = \sum_i \min(P_F(u_i|s), P_F(v_i|s))\mathbf{C}_{ii}'.
\tag{37}
$$

We obtain the closed-form formulation for optimal transport distance

$$
\begin{aligned}
\text{OT}(P_F(\cdot|s), P_F(\cdot|s')) = & \sum_{u \in \text{Child}(s)} P_F(u|s)\log(P_B(s|u)) + \mathbf{H}(P_F(\cdot|s')) \\
& + P_F(s'|s).(\log(P_B(s'|s)) + \log(P_F(s'|s))) \\
& + \sum_{i \in A_s^* \bigcap A_{s'}^*} \min(P_F(u_i|s), P_F(v_i|s'))c_i'.
\end{aligned}
\tag{38}
$$

**Lemma 1** *Given a squared diagonal cost matrix $\mathbf{C}'$ with non-positive entities in the diagonal, the solution of optimal transport problem between two distribution $P_F(\cdot|s)$ and $P_F(\cdot|s')$, which has the same number of support points, given cost matrix $\mathbf{C}'$ is given by:*

$$
\underset{\pi \in \Pi(P_F(\cdot|s), P_F(\cdot|s'))}{\min} \langle \mathbf{C}', \pi \rangle = \sum_i \min(P_F(u_i|s), P_F(v_i|s))\mathbf{C}_{ii}'.
\tag{39}
$$

**Proof of Lemma 1:** Let define

$$F(\pi) = \langle \mathbf{C}', \pi \rangle,$$

$$\overline{p}_{ij} = \begin{cases} \min(p_s^i, p_{s'}^i), & \text{if } i = j \\ \frac{\left(p_s^i - \min(p_s^i, p_{s'}^i)\right)\left(p_{s'}^j - \min(p_s^j, p_{s'}^j)\right)}{1 - \sum_k min(p_s^k, p_{s'}^k)} & \text{if } i \neq j. \end{cases}$$

where $p_s^i := P_F(u_i|s)$ and $p_{s'}^j := P_F(v_j|s')$.

We will prove that $\overline{\pi} \in \Pi\left(P_F(\cdot|s), P_F(\cdot|s')\right)$ and $F(\pi) \geq F(\overline{\pi}) \quad \forall \pi \in \Pi\left(P_F(\cdot|s), P_F(\cdot|s')\right)$.

It is not difficult to show that $\overline{\pi}_{ij} \geq 0$. From the definition of $\overline{\pi}$, we have

$$\sum_j^n \overline{\pi}_{ij} = \sum_{j \neq i} \overline{\pi}_{ij} + \overline{\pi}_{ii} = \sum_{j \neq i} \frac{\left(p_s^i - \min(p_s^i, p_{s'}^i)\right)\left(p_{s'}^j - \min(p_s^j, p_{s'}^j)\right)}{1 - \sum_k min(p_s^k, p_{s'}^k)} + \min(p_s^i, p_{s'}^i). \quad (40)$$

If $\min(p_s^i, p_{s'}^i) = p_s^i$ then

$$\sum_j^n \overline{\pi}_{ij} = 0 + \min(p_s^i, p_{s'}^i) = p_s^i. \quad (41)$$

else $\min(p_s^i, p_{s'}^i) = p_{s'}^i$ then

$$\sum_{j \neq i} \left(p_{s'}^j - \min(p_s^j, p_{s'}^j)\right) = \sum_j \left(p_{s'}^j - \min(p_s^j, p_{s'}^j)\right) = 1 - \sum_k \min(p_s^k, p_{s'}^k)$$

$$\implies \sum_j^n \overline{\pi}_{ij} = \left(p_s^i - \min(p_s^i, p_{s'}^i)\right) \frac{\sum_{j \neq i} \left(p_{s'}^j - \min(p_s^j, p_{s'}^j)\right)}{1 - \sum_k \min(p_s^k, p_{s'}^k)} + \min(p_s^i, p_{s'}^i) = p_s^i. \quad (42)$$

Therefore $\sum_j^n \overline{\pi}_{ij} = p_s^i = P_F(u_i|s)$. Similarly, $\sum_j^n \overline{\pi}_{ij} = p_s^j = P_F(v_j|s')$, combining with $\overline{\pi}_{ij} \geq 0$, we have

$$\overline{\pi} \in \Pi\left(P_F(\cdot|s), P_F(\cdot|s')\right). \quad (43)$$

Moreover

$$F(\pi) = \langle \mathbf{C}', \pi \rangle = \sum_i \pi_{ii} \mathbf{C}'_{\mathbf{ii}} \geq \sum_i \min(p_s^i, p_{s'}^i)\mathbf{C}'_{\mathbf{ii}} = \langle \mathbf{C}', \overline{\pi} \rangle = F(\overline{\pi}) \quad \forall \pi \in \Pi\left(P_F(\cdot|s), P_F(\cdot|s')\right). \quad (44)$$

As a consequence, we obtained the solution of optimal transport problem.

**closed-form solution for optimal transport distance at terminal state.** We will derive the closed-form solution for optimal transport distance in case of two neighbor states $s < s'$, in which $s'$ is a terminal state. In the case of Hyper-grid environment, EB-GFN experiments, and Biological Sequence Design, all terminal state $x$ have only one child that is the final state $s_f$, and $P_F(s_f|x) = 1 \quad \forall x$. Thus, the admissible couplings set $\prod(P_F(\cdot|s), P_F(\cdot|s'))$ has only one element. That is $\pi^* = P_F(\cdot|s)$. As a result, the optimal transportation distance between $P_F(.|s)$ and $P_F(.|s')$ is:

$$\text{OT}\left(P_F(\cdot|s), P_F(\cdot|s')\right) = \min_{\pi \in \prod(P_F(\cdot|s), P_F(\cdot|s'))} \langle \mathbf{C}, \pi \rangle = \langle \mathbf{C}, \pi^* \rangle. \quad (45)$$

Specially, in EB-GFN experiments, all children $u_i$ of $s$ is a terminal state so $d(u_i, s_f) = -\log(1) = 0$. This makes $\mathbf{C} = 0$ and $\text{OT}\left(P_F(\cdot|s), P_F(\cdot|s')\right) = 0$. In Hyper-grid environment experiment, for terminal sate $s'$ because $c_i' = 0$, we have:

$$\text{OT}\left(P_F(\cdot|s), P_F(\cdot|s')\right) = \sum_{u \in \text{Child}(s)} P_F(u|s) \log(P_B(s|u))$$

$$+ P_F(s'|s).(\log(P_B(s'|s)) + \log(P_F(s'|s))). \quad (46)$$

The Hyper-grid environment (Bengio et al., 2021a) (in section 4.1) and EB-GFN experiments (Zhang et al., 2022) (in section 4.3) satisfy two condition in Theorem 3.2. In Biological Sequence Design (Jain et al., 2022) (in section 4.2) such as protein and DNA sequences, the action space consists of actions adding a nucleic acid in $\{A, T, G, U\}$ and a amino acid respectively. Such settings satisfy former condition $a_i \neq a_k + a_h \quad \forall a_i, a_k, a_h \in \mathcal{A}$. However, the later condition $a_i + a_h = a_m + a_n, a_i \neq a_m \iff a_i = a_n, a_h = a_m, a_i \neq a_m$ is no longer true because the order property of action space, i.e, $a_i + a_j \neq a_j + a_i$. In this situation, the third terms in Eq. 14 is zero and we can still using the formulation in Eq. 14. Generally, the action space is independence and unique factorization.

## D.3 PROOF OF THEOREM C.1

By definition of the edge flow we have

$$\sum_{\tau:s\to s'\in\tau} P(\tau) = \sum_{\tau:s\to s'\in\tau} \frac{F(\tau)}{Z} = \frac{F(s\to s')}{Z} = P(s\to s'). \tag{47}$$

From that equation, we find that

$$
\begin{aligned}
\mathbb{E}_{\tau\sim\pi_{\boldsymbol{\theta}}}(\mathcal{L}_{\text{OT}}(\tau)) &= \mathbb{E}_{\tau\sim\pi_{\boldsymbol{\theta}}}\left(\sum_{s\to s'\in\tau} \text{OT}\left(P_F(\cdot|s), P_F(\cdot|s')\right)\right) \\
&= \sum_{\tau}\sum_{s\to s'\in\tau} \text{OT}\left(P_F(\cdot|s), P_F(\cdot|s')\right).P(\tau) \\
&= \sum_{s\to s'}\sum_{\tau:s\to s'\in\tau} \text{OT}\left(P_F(\cdot|s), P_F(\cdot|s')\right).P(\tau) \\
&= \sum_{s\to s'} \text{OT}\left(P_F(\cdot|s), P_F(\cdot|s')\right)\sum_{\tau:s\to s'\in\tau} P(\tau) \\
&= \sum_{s\to s'} \text{OT}\left(P_F(\cdot|s), P_F(\cdot|s')\right).P(s\to s') \\
&\propto \mathbb{E}_{s\to s'\sim\pi_{\boldsymbol{\theta}}}(\text{OT}\left(P_F(\cdot|s), P_F(\cdot|s')\right)).
\end{aligned}
\tag{48}
$$

$\square$

## E EXPERIMENT SETTINGS

In this part, we report experiment settings, including evaluation metrics for comparing the methods, hyper-parameter choices, and neural network architectures for all experiments. For biological sequence design tasks, we also give more details about the task description and datasets used for training. Note that the regularization coefficients provided in this part are task-specific. Specifically, $\lambda$ is chosen from a predefined set of values with different scales in each task. However, because all tasks in our experiment parts do not change the target distribution between the training and test time, the reported $\lambda$ is chosen to have the best model's performance.

### E.1 HYPER-GRID ENVIRONMENT

#### E.1.1 EVALUATION CRITERIA

To evaluate the performance, we measure the KL divergence between the actual and empirical distribution of the last $2\times 10^5$ visited states. The number of modes found during the training progress is also used to measure the learned models' performance.

#### E.1.2 IMPLEMENTATION DETAILS

**GFlowNet:** For the implementation of the GFlowNet model, we also follow the framework of Malkin et al. (2022): an MLP with two hidden layers of 256 dimensions each. The GFlowNet policy model, which includes both $P_F$ and $P_B$, is trained with a learning rate of 0.001 while the learning rate for total flow $Z_\theta$ is 0.1. We use a mini-batch size of 16 and 62500 training steps with the trajectory balance objective.

**Proposed OT regularization** The regularization coefficient is 0.02 for both Min OT, UB-OT, and Max OT in $4-D$ hypergrid environment and 0.1 for both Min OT, UB-OT, and Max OT in $8-D$ hypergrid environment.

### E.2 BIOLOGICAL SEQUENCE DESIGN

#### E.2.1 TASK DESCRIPTION & DATASETS

These experiments simulate the process of designing biological sequences, such as anti-microbial peptides, DNA, and protein sequences..., in drug discovery applications. This process often consists of an active loop with several rounds of ideating molecules and multiple-stage evaluations for filtering candidates, with rising levels of precision and cost. This characteristic makes the diversity of proposed candidates a considerable concern in the ideation phase because many similar candidates can all fail in the later phases.

Specifically, we consider the problem of finding objects $x$ in the space of discrete objects $\mathcal{X}$, that maximize a given oracle $f : \mathcal{X} \mapsto \mathbb{R}^+$. Here, we can only query this oracle $N$ times, each with an input batch of fixed size $b$. This can form $N$ rounds of evaluation in the active learning setting, where the generative policy is initially given a dataset $D_0 = \left\{ \left(x_1^0, y_1^0\right), \ldots, \left(x_n^0, y_n^0\right)\right\}$ collected from the oracle, where $y_i^0 = f(x_i^0)$ for $1 \leq i \leq n$.

Because the oracle can only be called limited, we also train a supervised proxy model $M$ that predicts $y$ from $x$ to approximate the oracle $f$. Specifically, in $i$-th round, given the current dataset $\mathcal{D}_i$, this proxy model can be used as a reward function $R$ to collect additional observations to train our generative policy to propose a batch of candidates $\mathcal{B}_i = \left\{ x_1^i, \ldots, x_b^i \right\}$. Then the current dataset $\mathcal{D}_i$ is updated for the next round of evaluation as $\mathcal{D}_{i+1} = \mathcal{D}_i \cup \left\{ \left(x_1^i, y_1^i\right), \ldots, \left(x_b^i, y_b^i\right)\right\}$ where $y_j^i = f\left(x_j^i\right)$.

Following the framework of Jain et al. (2022), we will conduct experiments on the biological sequence design tasks:

**Anti-Microbial Peptide Design:** This task aims to generate short amino-acid sequences of length lower than 51, which have anti-microbial properties. The vocabulary has 20 amino-acids $[A, C, D, E, F, G, H, I, K, L, M, N, P, Q, R, S, T, V, W, Y]$. The active learning algorithm is evaluated for $N = 10$ rounds, with the number of candidates generated each round $b = 1000$. The initial dataset $\mathcal{D}_0$ contains 3219 AMPs and 4611 non-AMP sequences, which is collected from the DBAASP database Pirtskhalava et al. (2021).

**TFBind 8:** The goal of this task is to generate DNA sequences of length 8, which have high binding activity with human transcription factors. The vocabulary has 4 nucleobases $[A, C, T, G]$. The active learning algorithm is evaluated for $N = 10$ round, with the number of candidates generated each round $b = 128$. The initial dataset $\mathcal{D}_0$ contains $32,898$ samples, which is half of all possible DNA sequences of length 8 having lower scores. The data and the oracle used are from Barrera et al. (2016).

**GFP:** The objective of this task is to generate protein sequences of length 237 that have high fluorescence. The vocabulary is similar to the one of the AMP task (size 20). The active learning algorithm is evaluated for $N = 10$ round, with the number of candidates generated each round $b = 128$. The initial dataset $\mathcal{D}_0$ contains $5,000$ samples, which is from Rao et al. (2019); Sarkisyan et al. (2016) together with the oracle.

### E.2.2 EVALUATION CRITERIA

To evaluate the performance, we also use the metrics as in Jain et al. (2022). Specifically, considering a set of candidates $\mathcal{D}$, we have the following metrics:

**Performance score:** mean score of the candidates in the set

$$\text{Mean}(\mathcal{D}) = \frac{\sum_{(x_i, y_i) \in \mathcal{D}} y_i}{|\mathcal{D}|}, \tag{49}$$

**Diversity:** a measurement of how well the generated candidates can capture the modes of the distribution implied by the oracle

$$\text{Diversity}(\mathcal{D}) = \frac{\sum_{(x_i, y_i) \in \mathcal{D}} \sum_{(x_j, y_j) \in \mathcal{D} \setminus \{(x_i, y_i)\}} d\left(x_i, x_j\right)}{|\mathcal{D}|(|\mathcal{D}| - 1)}, \tag{50}$$

where $d$ is a distance defined over $\mathcal{X}$, such as Levenshtein distance Miller et al. (2009).

**Novelty:** a measure of the difference between the candidates in $\mathcal{D}$ and $\mathcal{D}_0$

$$\text{Novelty}(\mathcal{D}) = \frac{\sum_{(x_i, y_i) \in \mathcal{D}} \min_{s_j \in \mathcal{D}_0} d\left(x_i, s_j\right)}{|\mathcal{D}|}. \tag{51}$$

These metrics will be evaluated on the set of candidates that have top $K$ scores $\mathcal{D} = \text{TopK}\left(\mathcal{D}_N \setminus \mathcal{D}_0\right)$.

### E.2.3 IMPLEMENTATION DETAILS

For the implementation of the GFlowNet-AL baseline model, we use the previously published implementation with slight changes, which follows the training setups of Jain et al. (2022):

**Proxy model**: We parameterize it as an MLP with two hidden layers, each having 2048 hidden units, and use ReLU activation. We also use ensembles of 5 models with same architecture for uncertainty estimation. For the acquisition function, we use UCB ($\mu + \kappa\sigma$) with $\kappa = 0.1$. The proxy is trained with MSE loss using mini-batch of size 256 and Adam optimizer with $(\beta_0, \beta_1) = (0.9, 0.999)$ and learning rate $10^{-4}$. During training, early stopping is also used by evaluating the validation set containing $10\%$ of the data.

**GFlowNet generator**: We use an MLP with 2 hidden layers of 2048 hidden units each. The model is trained with trajectory balance objective as the main loss function, by using Adam optimizer with $(\beta_0, \beta_1) = (0.9, 0.999)$. Additionally, $\log Z$ is trained with a learning rate of $10^{-3}$ for AMP, TF Bind 8 task, and $5 \times 10^{-3}$ for GFP task. Other hyper-parameters are shown in the following table:

| Hyper-parameter | AMP | TF Bind 8 | GFP |
|---|---|---|---|
| $\delta$ : Uniform Policy Coefficient | 0.001 | 0.001 | 0.05 |
| Learning rate | $5 \times 10^{-4}$ | $10^{-5}$ | $10^{-3}$ |
| $m$ : Minibatch size | 32 | 32 | 32 |
| $\beta$ : Reward Exponent $R(x)^\beta$ | 3 | 3 | 3 |
| T : Training steps | 10,000 | 5,000 | 50,000 |

Table 5: Hyper-parameters for the GFlowNet.

There are some changes in hyper-parameter choices and the number of active learning rounds in the TF Bind 8 task and the GFP task compared to the original training setups of Jain et al. (2022). However, during the experiment, we observed that these settings helped us get the closest results to the reported one in Jain et al. (2022).

**Proposed OT regularization** The regularization coefficients for Min OT, UB OT, and Max OT are the same for each biological sequence design task. Specifically, the coefficients for the AMP, TF Bind 8, and GFP task are 0.025, 0.1, and 0.02 correspondingly.

### E.3 SYNTHETIC DISCRETE PROBABILISTIC MODELING TASKS

#### E.3.1 EVALUATION CRITERIA

To evaluate the performance, we keep the same evaluation criteria in Zhang et al. (2022), where they use the NLL of a large independent sample of ground truth data and the exponential Hamming MMD (Gretton et al. (2012)) between ground truth data and generated samples as performance metrics. To measure NNL and MMD, we use 10 fixed sets, and each set consists of 4000 ground truth data samples.

#### E.3.2 IMPLEMENTATION DETAILS

**GFlowNet:** For the implementation of the GFlowNet model, we use an MLP with 2 hidden layers of 512 dimensions each. The GFlowNet policy model, which includes both $P_F$ and $P_B$, is trained with a learning rate of 0.001. We use a mini-batch size of 128 and $1e5$ training steps with the trajectory balance objective.

**EBMs:** For the implementation of the Energy-Based Model, we use an MLP with 3 hidden layers of 256 dimensions each. The learning rate is 0.001.

**Proposed OT regularization**: The regularization coefficient is 0.001 for both Min OT and UB OT and is the same for all tasks.

## F ADDITIONAL EXPERIMENT RESULTS

### F.1 ABLATION STUDY ABOUT VARYING $\lambda$

Specifically, we will further investigate the proposed path regularization via OT with different values of the regularization coefficient $\lambda$ in the 8-D hyper-grid environment in Section 4.1. In addition, the regularization coefficient is selected from the set $(0.001, 0.01, 0.1, 0.4)$. We plot the mean results over 10 runs for each configuration in Fig. 3.

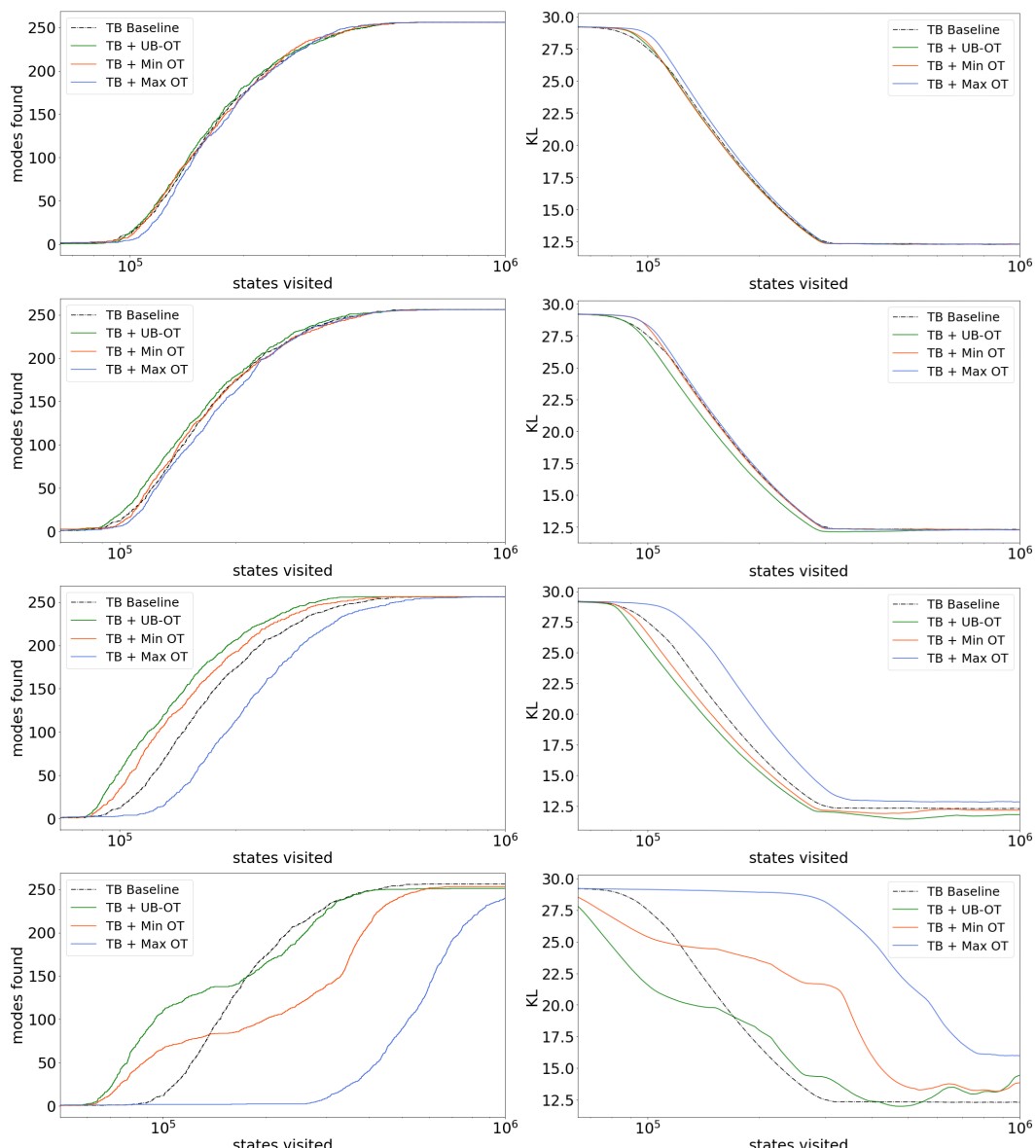

Figure 3: Results on the $8 - D$ hyper-grid environment with $\lambda \in (0.001, 0.01, 0.1, 0.4)$ (from top to bottom). Left: Number of modes found during training. Right: KL divergence between the true and empirical distribution.

Note that the good range of values for the regularization coefficient is observed to highly depend on the specific setting of the experiment task. Here, we can see that when $\lambda$ is relatively small, such as $\lambda \in (0.001, 0.01)$, the performance of GFlowNets trained additionally with our proposed regularization via 0T does not seem to be significantly different from the baseline model's performance, which holds for both UB OT, Min OT, and Max OT. This may be resulted from the small contribution of regularization to the regularized training objective, which is caused by the not large enough value of $\lambda$. Specifically, when $\lambda = 0.01$, we can still see that the performance of GFlowNets trained by minimizing the upper bound is slightly better than the baseline's result.

In addition, when $\lambda$ is relatively large ($\lambda = 0.4$), the result of learned GFlowNets is even worse than the baseline, which may be due to the large value of $\lambda$ forces the model's learning focus on the regularization part more than necessary, which badly affects the optimization of the main training objective (trajectory balance objective). Specifically, this can be observed in the lower KL divergence of both UB OT, Min OT, and Max OT compared to the baseline.

Meanwhile, when $\lambda = 0.1$, GFlowNets trained by minimizing the OT regularization and its upper bound clearly perform better than the baselines regarding the number of modes found and KL divergence between the actual and empirical distribution, which proves our motivation that minimizing the proposed path regularization is more beneficial in this circumstances.

### F.2 ADDITIONAL RESULTS OF THE HYPERGRID ENVIRONMENT

We also plot the mean results over 10 runs for each configuration with variance in Fig. 4.

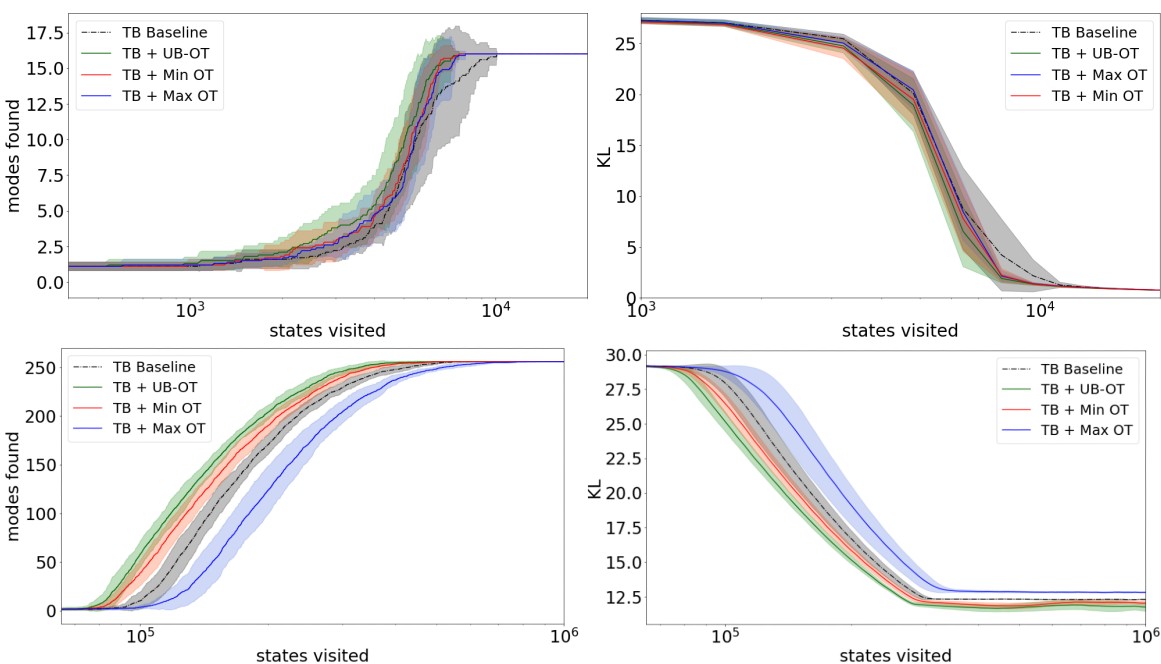

Figure 4: Results with variance on the $4 - D$ (upper) and $8 - D$ (lower) hyper-grid environment. Left: Number of modes found during training. Right: KL divergence between the true and empirical distribution.

