# OpenReview forum: "Improving Generative Flow Networks with Path Regularization"
_ICLR.cc/2023/Conference — Submitted to ICLR 2023_

### Official Review · Reviewer_86A8 · 2022-10-19

**Confidence:** 4
**Correctness:** 3
**Technical Novelty And Significance:** 3
**Empirical Novelty And Significance:** 2
**Recommendation:** 5

**Clarity, Quality, Novelty And Reproducibility:**

### Clarity & Quality
This paper is OK w.r.t. readability. But I think it would be better to move the content of Sec. 3.3 to the end of Sec. 3.1 to give a clear demonstration of the specific algorithm.

### Novelty
The idea of this paper is novel.

### Reproducibility
The corresponding code implementation is provided.

**Strength And Weaknesses:**

## Strengths
The proposal of using optimal transport theory to regularize GFlowNet training is novel and interesting. Three different regularization terms are proposed accordingly.
1. The authors propose the definition of directed distance given the "back-and-forth trajectory" idea in EB-GFN paper.
2. Tractable calculation methods are derived with theoretical insights to achieve the proposed algorithms.
3. A series of experiments are conduct to verify the advantage of the path regularization method.

## Weaknesses
1. In Sec. 1, it is written that the motivation of minimizing path regularization is to enhance flow on high flow trajectories. This is true, but what I do not understand is the connection with generalization. First, GFlowNet is a way to amortize the MCMC sampling problem, and since there is no change of the target distribution between the training and test time, I do not think "generalization" is a proper notion to discuss the performance of GFlowNet. Second, such motivation is contrary to the one of GFlowNet, who aims to sample *proportional* to a given reward function, rather than finding the high-score objects. Such difference is the core motivation for why people need GFlowNets instead of directly using reinforcement learning or other related methods.

2. In Sec. 1 and Sec. 3.1, the authors also say maximizing the path regularization term gives more diverse effects. I agree with this point, but since minimizing and maximizing the path regularization term gives different regularization effects, I wonder which effect is needed by GFlowNet? It seems the authors only simply testify the two methods (Min OT & Max OT) without further analysis. Maybe different tasks would require different behaviors (and thus need different regularization effects)? For example, in the toy hypergrid experiment, if we make the reward function to be more sparse, would the Max OT method show more benefit? I think it would be great if the author could provide more analysis on this point.

3. Ablation about the hyperparameter $\lambda$, namely the coefficient of the regularization term, is missing. It would be great to have a plot / table of the performance of different methods with different values of $\lambda$ (e.g. both positive and negative values) on hypergrid (since it is the easiest task, but it would be better to have ablation results on other tasks).

4. The forward and backward policy are in symmetric positions, which says, either of them is able to determine the whole GFlowNet distribution. If the method proposed in this work is effective, then one thing worth trying is to put a similar technique to regularize the distance of backward policy on intermediate states.

5. The result of "Max OT" is missing in the probability modeling tasks. Is there any particular reason?

### Minors
1. In Sec. 4.3, there is a gap between the MMD values in EB-GFN paper and this paper. I feel this is about the number of samples used to calculate the MMD. Theoretically, MMD value should always be positive. However, it could be negative if estimated with a small number of samples, for example in original ALOE paper they use a fixed 4000 samples. In EB-GFN paper, the MMD is calculated with 10 times average, each time with a different set of 4000 samples (not fixed samples), and the results are positive.  It is OK to compare with a consistent metric like in the current paper, but it would be great if the numbers of this part could be re-calculated for better reliability.

2. One of the advantage of Wasserstein distance is that it does not require the distributions to have the same support, as mentioned in Sec. 1. In molecule tasks, this unequal support would really matter as the action space is more structured in graph generation. In this sense, the molecule task should be a better playground for the proposed method (otherwise, why not directly use KL or other simple divergences? They are much easier to compute than OT).

3. There are a couple of typos and misuses of notations.

**Summary Of The Paper:**

This work proposes three different ways to regularize GFlowNet training with Wasserstein distance. Implementations of the regularization term and an upper bound of it are derived for tractable training. The path regularization is claimed to be capable of generating more diverse candidates via maximizing the Wasserstein distance or improve the generalization by minimizing the Wasserstein distance. The proposed methods are evaluated on a series of GFlowNet-related tasks, including hypergrid, biological sequence design and synthetic probabilistic modeling.

**Summary Of The Review:**

This work is with reasonable writing and experiments. However, I have some concerns about the motivation, and also feel more experiments could be conduct to amplify the story of this paper. I would definitely consider raising the score if some of the main weaknesses could be reasonably addressed :)

---

> ### Author Response · Authors · 2022-11-18
> **Author's Response to Reviewer 86A8 (1)**
>
> ### Response:
>
> Thank you for your thoughtful review and valuable feedback. Below we address your concerns.
>
> **Q1**. In Sec. 1, it is written that the motivation of minimizing path regularization is to enhance flow on high flow trajectories. This is true, but what I do not understand is the connection with generalization. First, GFlowNet is a way to amortize the MCMC sampling problem, and since there is no change of the target distribution between the training and test time, I do not think "generalization" is a proper notion to discuss the performance of GFlowNet. Second, such motivation is contrary to the one of GFlowNet, who aims to sample proportional to a given reward function, rather than finding the high-score objects. Such difference is the core motivation for why people need GFlowNets instead of directly using reinforcement learning or other related methods.
>
> **Reply**:
> * First, let us explain more clearly the notion of "generalization" in the context of GFlowNets. Followed by [1], GFLowNets was initially formulated as a method of amortizing the MCMC problem. Moreover, while mode mixing problems seriously challenge MCMC methods, GFlowNets can alleviate these problems ([1], [2], [3]). Specifically, GFlowNets can guess and sample modes that haven't been visited yet, if there exist learnable patterns in the target distribution ([2]), which represents the generalization capability of GFlowNets. In addition, this generalization ability is also stated in [3], as the parametric policy of GFlowNets can give inductive biases and allow it to generalize to the unseen states during training progress.
> * Second, the motivation, that minimizing the proposed path regularization can improve generalization ability of GFlownets, can be summarized as follows:
>    * In specific problems like probabilistic modeling in discrete spaces or modeling a synthetic hypergrid environment, we want to improve the generalization ability of GFlowNet. In other words, we expect the learned model can better discover the latent structure of the target distribution, i.e. (1) In the synthetic hypergrid environment, there are only states with high rewards near its corners, or (2) When modeling image datasets in discrete spaces, the modes, which have many zero entries, only concentrate on small regions of the space.
>    * In addition, we can have many different learned GFlowNets for solving one typical problem, i.e., in the hypergrid experiment, the forward transition probability of each available action at each state can be almost equal or focus on some specific actions in the action space (Figure 2 in [3]).
>    * However, in the above problems, it would be more beneficial to learn GFlowNets such that the forward policy at each state will have the focused tendency, as it will help the model better discover the latent structure of the target distribution (the results in Section 4.1 also show these effects). Specifically, our regularization can be interpreted as placing a prior constraint that forces the forward policies of two neighbor states to be similar in the way that they both have the focused tendency of choosing the following action, which will enhance the flow on high flow trajectories. This constraint is optimized together with the primary learning objective of GFlowNets, usually with a small regularization coefficient, which will not be contradicted.
>
> **References**:
>
> [1]: Emmanuel Bengio, Moksh Jain, Maksym Korablyov, Doina Precup, and Yoshua Bengio. Flow network based generative models for non-iterative diverse candidate generation. NeurIPS 2021.
>
> [2]: Dinghuai Zhang, Nikolay Malkin, Zhen Liu, Alexandra Volokhova, Aaron C. Courville, and Yoshua Bengio. Generative flow networks for discrete probabilistic modeling. ICML 2022.
>
> [3]: Nikolay Malkin, Moksh Jain, Emmanuel Bengio, Chen Sun, and Yoshua Bengio. Trajectory balance: Improved credit assignment in gflownets. NeurIPS 2022.

---

> > ### Author Response · Authors · 2022-11-18
> > **Author's Response to Reviewer 86A8 (2)**
> >
> > **Q2**. In Sec. 1 and Sec. 3.1, the authors also say maximizing the path regularization term gives more diverse effects. I agree with this point, but since minimizing and maximizing the path regularization term gives different regularization effects, I wonder which effect is needed by GFlowNet? It seems the authors only simply testify the two methods (Min OT \& Max OT) without further analysis. Maybe different tasks would require different behaviors (and thus need different regularization effects)? For example, in the toy hypergrid experiment, if we make the reward function to be sparser, would the Max OT method show more benefit? I think it would be great if the author could provide more analysis on this point.
> >
> > **Reply**:  We agree with your opinion that different tasks would require different regularization effects. In fact, it has been stated when we introduce our motivation and also concluded again in the first paragraph of Section 4 with specific tasks for illustrating the regularization effects. Let us make this point more clearly:
> > * In specific problems like probabilistic modeling in discrete spaces or modeling a synthetic hypergrid environment, the states with high rewards only concentrate on small regions of the whole object space. Then in these cases, minimizing the proposed regularization can be more beneficial by making the forward policy at each state only focus on some actions in the action space, which implicitly forces GFlowNets to find states with high rewards rather than exploring the environment. For example, in Section 4.1, we have demonstrated that the learned model by minimizing the regularization finds the modes faster and achieves some level of low KL error more quickly, which indicates minimizing the regularization help the learned model better discover the latent structure of the target distribution.
> > * On the other hand, in the multi-round active learning settings, such as molecule generation or biological sequence design, the diversity and novelty of the generated samples become the most crucial concern of GFlowNets ([1], [2], [3]), such that the generated candidates are expected to be good enough to survive in the later evaluation phases. Moreover, in these settings, the rewards for learning GFlowNets are also provided by a supervised approximating proxy because of limited access to the true oracle. However, because the approximating proxy is quite unreliable, diverse candidates will be helpful. In addition, diverse candidates also provide better knowledge to update the approximating proxy. Then in this circumstance, maximizing the proposed regularization can help the learned GFlowNets perform better in generating more diverse and novel samples.
> >
> > **References**:
> >
> > [1]: Emmanuel Bengio, Moksh Jain, Maksym Korablyov, Doina Precup, and Yoshua Bengio. Flow network based generative models for non-iterative diverse candidate generation. NeurIPS 2021.
> >
> > [2]: Nikolay Malkin, Moksh Jain, Emmanuel Bengio, Chen Sun, and Yoshua Bengio. Trajectory balance: Improved credit assignment in gflownets. NeurIPS 2022.
> >
> > [3]: Moksh Jain, Emmanuel Bengio, Alex Hern ́andez-Garc ́ıa, Jarrid Rector-Brooks, Bonaventure F. P.Dossou, Chanakya Ajit Ekbote, Jie Fu, Tianyu Zhang, Michael Kilgour, Dinghuai Zhang, Lena Simine, Payel Das, and Yoshua Bengio. Biological sequence design with gflownets. ICML 2022.
> >
> > ----
> >
> > **Q3**.  The forward and backward policy are in symmetric positions, which says, either of them is able to determine the whole GFlowNet distribution. If the method proposed in this work is effective, then one thing worth trying is to put a similar technique to regularize the distance of backward policy on intermediate states.
> >
> > **Reply**:  Indeed, we are motivated initially to derive a regularization method that works on any generic structure of the underlying directed acyclic graph $(\mathcal{S}, \mathcal{A})$. However, when the given DAG is a tree, such as in biological sequence design tasks, a unique trajectory exists from the source state to any state. In other words, in that case, $P_{B}(s'|s)$ always equals $1$ when $s \rightarrow s' \in \mathcal{A}$, which results in the regularization on backward policy doesn't make sense.
> >
> > By the way, we agree that it's interesting to try to derive a similar method by replacing the forward policy with the backward policy, maybe on another problem having underlying DAG that is not a tree. With similar techniques as in Theorem 3.1 and 3.2, we can derive the formulations of the upper bound and closed form solution for the OT distance. But it will take time to analyze whether the formulation has meanings like our proposed method, which have been investigated in Section 3.1 and 3.2, and also run additional experiments to demonstrate its effect if the previous part holds.

---

> > > ### Author Response · Authors · 2022-11-18
> > > **Author's Response to Reviewer 86A8 (3)**
> > >
> > > **Q4**. Ablation about the hyperparameter $\lambda$, namely the coefficient of the regularization term, is missing. It would be great to have a plot / table of the performance of different methods with different values of $\lambda$(e.g. both positive and negative values) on hypergrid (since it is the easiest task, but it would be better to have ablation results on other tasks).
> > >
> > > **Reply**:  We will further investigate the proposed path regularization via OT with different values of the regularization coefficient $\lambda$ in the hypergrid environment in Section 4.1. While details about the model's architecture, hyper-parameters, and evaluation criteria are the same as those provided in Appendix E, we will study a sparser hypergrid environment with the number of dimensions $N=8$ and the side length $H=8$ to have a better insight about the effect of the proposed regularization method and the regularization coefficient $\lambda$. In addition, the regularization coefficient is selected from the set $(0.001, 0.01, 0.1, 0.4)$. We plot the mean results over $10$ runs for each configuration in Fig. 1 here: https://sites.google.com/view/improve-gflownets-ot-rebuttal.
> > > * Note that the good range of values for the regularization coefficient is observed to highly depend on the specific setting of the experiment task. Here, we can see that when $\lambda$ is relatively small, such as $\lambda \in (0.001, 0.01)$, the performance of GFlowNets trained additionally with our proposed regularization via 0T does not seem to be significantly different from the baseline model's performance, which holds for both UB OT, Min OT, and Max OT. This may be resulted from the small contribution of regularization to the regularized training objective, which is caused by the not large enough value of $\lambda$. Specifically, when $\lambda = 0.01$, we can still see that the performance of GFlowNets trained by minimizing the upper bound is slightly better than the baseline's result.
> > > * In addition, when $\lambda$ is relatively large ($\lambda = 0.4$), the result of learned GFlowNets is even worse than the baseline, which may be due to the large value of $\lambda$ forces the model's learning focus on the regularization part more than necessary, which badly affects the optimization of the main training objective (trajectory balance objective). Specifically, this can be observed in the lower KL divergence of both UB OT, Min OT, and Max OT compared to the baseline.
> > > * Meanwhile, when $\lambda = 0.1$, GFlowNets trained by minimizing the OT regularization and its upper bound clearly perform better than the baselines regarding the number of modes found and KL divergence between the actual and empirical distribution, which proves our motivation that minimizing the proposed path regularization is more beneficial in this circumstances. Otherwise, Max OT seems unsuitable because of its motivation to improve the model's exploration, while we only need the high-reward states near the corners, and the majority of states have minimal rewards.
> > > * As a result, we will replace the results on the $7-D$ hypergrid environment with the $8-D$ one to demonstrate a better effect of our method.
> > >
> > > ----
> > >
> > > **Q5**.  In Sec. 4.3, there is a gap between the MMD values in EB-GFN paper and this paper. I feel this is about the number of samples used to calculate the MMD. Theoretically, MMD value should always be positive. However, it could be negative if estimated with a small number of samples, for example in original ALOE paper they use a fixed 4000 samples. In EB-GFN paper, the MMD is calculated with 10 times average, each time with a different set of 4000 samples (not fixed samples), and the results are positive. It is OK to compare with a consistent metric like in the current paper, but it would be great if the numbers of this part could be re-calculated for better reliability.
> > >
> > > **Reply**: To compute MMD and NLL values, we use 10 different sets of 4000 true samples; each set was generated by a true oracle with seeds from 1 to 10, and reported the 10 times average results. In EB-GFN, [1] suggests using 10 sets of 4000 true samples to reduce the variance and archive positive results of MMD. We keep the setting the same as in [1]. However, there is a gap between them, and some results of MMD are negative. So we only use our reproduced results of EB-GFN to compare with Min OT, UB, and Max OT.
> > >
> > > **References**:
> > >
> > > [1]: Dinghuai Zhang, Nikolay Malkin, Zhen Liu, Alexandra Volokhova, Aaron C. Courville, and Yoshua Bengio. Generative flow networks for discrete probabilistic modeling. ICML 2022.

---

> > > > ### Author Response · Authors · 2022-11-18
> > > > **Author's Response to Reviewer 86A8 (4)**
> > > >
> > > > **Q6**.  The result of "Max OT" is missing in the probability modeling tasks. Is there any particular reason?
> > > >
> > > > **Reply**: Our paper has updated the Max-OT results for synthetic discrete probabilistic modeling tasks (Synthetic EB-GFN) in Table 4. Specifically, when reproducing the baseline of GFlowNets, we use the same settings as in [1]. However, there is a gap between them. To highlight the best result, we will not consider the results reported in the EB-GFN paper but still report them. We only take into account the the reproduce results of EB-GFN when comparing with our methods (Min OT, Max OT, and UB OT).
> > > > | Metric| Method            |2spirals     |8gaussians    | circles    |moons      |pinwheel  |swissroll     | checkerboard   |
> > > > |-|-|-|-|-|-|-|-|-|
> > > > | NLL   | PCD               |20.094       |19.991    | 20.565    | 19.763    | 19.593   |20.172    |21.214         |
> > > > |       |ALOE               |20.295    | 20.350   | 20.565    | 19.287    | 19.821   |20.160    |54.653         |
> > > > |       |ALOE +             |20.062    |19.984    |20.570     | 19.743    | 19.576   | 20.170   | 21.142        |
> > > > |       |EB-GFN (paper)     |20.050    |19.982    |20.546     |19.732     |19.554    |20.146    |20.696         |
> > > > |       |EB-GFN             |20.0679   |19.9862   |**20.5598**   | 19.7324    |19.5735   |20.1599   |20.6839        |
> > > > |       |EB-GFN + Max OT    |20.0673   |19.9857   | 20.5599   |19.7319    |19.5714   |20.1597   |20.6837        |
> > > > |       |EB-GFN + UB OT     |20.0651   |**19.9854**  | 20.5600    |**19.7305**   | 19.5707   |20.1596   |20.6836        |
> > > > |       |EB-GFN + Min OT    |**20.0640**  | 19.9855   |**20.5598**   | 19.7308    |**19.5699**  | **20.1595**  | **20.6831**   |
> > > > | MMD   | PCD               |2.160     | 0.954    |0.188      | 0.962     | 0.505    | 1.382    |2.831          |
> > > > |       |ALOE               |21.926    | 107.320  | 0.497     | 26.894    | 39.091   | 0.471    | 61.562        |
> > > > |       |ALOE +             |0.149     |0.078     | 0.636     |0.516      |1.746     | 0.718    |12.138         |
> > > > |       |EB-GFN (paper)     |0.583     |0.531     |0.305      |0.121      |0.492     |0.274     |1.206          |
> > > > |       |EB-GFN             |0.3012    |0.0408    |-0.1724    |-0.1744    |0.2056    |0.1555    |-0.0986        |
> > > > |       |EB-GFN + Max OT    |0.3258    |0.0197    |-0.1919    |-0.0456    |0.1377    |0.0763    |-0.0903        |
> > > > |       |EB-GFN + UB OT     |0.2902    |**0.0102**   | **-0.2819**   | -0.1253    |0.1561    |**0.0257**   | -0.0923        |
> > > > |       |EB-GFN + Min OT    |**0.1816**   | 0.0343    |-0.2775    |**-0.1966**   | **0.1220**   | 0.1334    |**-0.1071**   |
> > > >
> > > > **References**:
> > > >
> > > > [1]: Dinghuai Zhang, Nikolay Malkin, Zhen Liu, Alexandra Volokhova, Aaron C. Courville, and Yoshua Bengio. Generative flow networks for discrete probabilistic modeling. ICML 2022.
> > > >
> > > > ----
> > > >
> > > > **Q7**. One of the advantage of Wasserstein distance is that it does not require the distributions to have the same support, as mentioned in Sec. 1. In molecule tasks, this unequal support would really matter as the action space is more structured in graph generation. In this sense, the molecule task should be a better playground for the proposed method (otherwise, why not directly use KL or other simple divergences? ).
> > > >
> > > > **Reply**:  We agree that the motivation for using OT is that it does not require two interested distributions to share the same set of supports. Additionally, in our paper, we consider the forward policy at each state as a discrete measure whose set of supports is precisely the set of its child states, which implies that the set of supports of the forward policies of two neighbor states are not the same. Then the proposed method makes sense in the problems solved by GFlowNets, at least all problems in our experiment part. Another reason is that the cost used in our OT distance can capture the given DAG's structure and the GFlowNet's flow. Meanwhile, directly using KL divergence cannot take into account the given DAG's structure.
> > > >
> > > > Indeed, another view exists that the set of supports of the forward policy at each state is the set of allowed actions. However, this view has its drawbacks that there exist many cases that when $\text{action}(s') \subset \text{action}(s)$ then $D_{KL}(P_F(\cdot|  s)||P_F(\cdot|s'))$ is infinite, which will not be meaningful. Note that these cases frequently exist in all tasks in our experiment parts, especially hypergrid environment or discrete probabilistic modeling tasks. Moreover, minimizing or maximizing KL divergence will only affect the two interested distributions in terms of their values, which intuitively can not have the effects in Section 3.1.
> > > >
> > > > ----
> > > >
> > > > We hope we have cleared your concerns about our work. We have also revised our manuscript according to your comments, and we would appreciate it if we can get your further feedback at your earliest convenience.

---

> > > > > ### Author Response · Authors · 2022-11-24
> > > > > **Response to Reviewer 86A8 - Any further questions on our current draft**
> > > > >
> > > > > We would like to thank you again for your thoughtful reviews and valuable feedback.
> > > > >
> > > > > We would appreciate it if you could let us know if our responses have addressed your concerns and whether you still have any other questions on the current draft and our rebuttal.
> > > > >
> > > > > We would be happy to do any follow-up discussion or address any additional comments.

---

### Official Review · Reviewer_ayzy · 2022-10-24

**Confidence:** 5
**Correctness:** 3
**Technical Novelty And Significance:** 3
**Empirical Novelty And Significance:** 2
**Recommendation:** 5

**Clarity, Quality, Novelty And Reproducibility:**

Quality and clarity: The exposition is good, but could give better attribution to past work (see above). Experiments are unconvincing and lack interpretation.

Originality: Good. This is the first attempt to answer the question of which solution to the GFlowNet policy optimization problem -- which generally has a high-dimensional family of global minima -- should be chosen.

Reproducibility: Good. Code is provided, based on code from past work (though I did not run it myself).

**Strength And Weaknesses:**

Comments on experiments:
- (++) Breadth of experiments: there is hypergrid, sequence design, energy-based modeling -- which all use different codebases in the original implementations from the papers that introduced them. I appreciate the engineering work and the variety of applications considered (simple reward matching, active learning, energy-based modeling).
- (--) Many of the results are unconvincing.
  - For the hypergrid environment, the difference between algorithms is almost invisible, and it could be compensated by small changes in learning rate. I would suggest trying  larger values of H and D, where sparsity may become more important, and showing more clearly the dependence on the most important hyperparameters (regularization weight -- how was it chosen?).
  - In the EB-GFN experiments, there is practically no difference between the proposed algorithm and baselines. Furthermore, the values highlighted in Table 4 are sometimes not actually the best one in their columns, which is misleading.
- (+/-) It is interesting that the Max OT approach performs best in the active learning settings. What is the interpretation of this result? Are more entropic policies beneficial in the AL setting? This is a potential strong point that should be investigated further.
- (-) In practice, what is the computational overhead of UB and Min/Max OT?
  - There are simpler (and less computationally expensive) ways to regularize a GFlowNet to either encourage or discourage sparsity. The simplest is just to penalize the entropy of the backward policy. It would be interesting to see how these compare to the OT-motivated regularization proposed here.

Major comments on writing and theory:
- (++) The motivation for regularizing the GFlowNet policy makes sense and, as far as I can tell, the math is right.
- (-) Large parts of the background on GFlowNets, especially Appendix B, closely follow or even copy [Malkin et al., 2022], abbreviated [TB], in the flow of exposition and in the notation. For example, the first sentence of "Learning GFlowNets" (p.14) is identical to the first sentence of 2.2 in [TB], and all choices of notation are the same. Borrowing the structure and notation is fine, but it should be attributed, with a sentence such as "this exposition closely follows that of ...".
  - In addition, in the section "Learning GFlowNets" on p.3, the proof of correctness of TB is misattributed (in fact, it appears in [TB], not [Bengio et al.]).
- (-) In Definition 3.1, the distance between states is only defined for states that are ancestors of one another. (By the way, in that definition, one should not call a sequence of actions a "transition".) However, on p.4, it is generalized to distances between any pair of states.
  - This generalization should be explicitly stated, and it should also be stated in what sense the resulting d(s,s') is a "distance". For example, it is not symmetric, and it is possible that the distance between two distinct states is 0 if the transition probability between them is 1. Thus all we can say about d is that it is a pseudoquasimetric (since it is nonnegative, d(s,s)=0, and it satisfies the triangle inequality).
  - The cost matrix in equation (9) is only an approximation to the true distance, since the true distance may be the NLL of a trajectory with more than three edges and may not just go "back" and then "forth". This should also be stated.

Minor comments:
- This sentence on p.2 does not make sense to me: "Compared with KL divergence, the biggest problem of KL divergence is that it was infinite for a variety of distributions with unequal support". Does this mean that KL divergence cannot be used as the distance metric because KL(p||q) is infinite if (support of p)-(support of q) has nonzero measure under p?
- The "pseudo backward policy" requires some more details. It is not actually a conditional distribution (may not sum to 1). So what exactly is meant by the cross-entropy in (17)?
- In (19), $\pi_\theta$ was not defined. (Is it the untempered forward policy?)
- The paper would benefit from some proofreading for small writing bugs, which nonetheless do not interfere with understanding of the paper. Here are a few:
  - Inconsistent capitalization and grammatical number in the name of the algorithm throughout the text, such as  "GFlownet", "GflowNet", "GFLowNet" instead of "GFlowNet". GFlowNet is singular and GFlowNets are plural.
  - Similar comment about "trajectory balance objective" (sometimes called "trajectory balanced objective" and inconsistently capitalized)
  - "natural interpreted" -> "naturally interpreted" (p.2)
  - "shortage path" -> "shortest path" (several places)
  - "can be define" -> "can be defined" (p.4)
  - Some equations are not in math mode (p.6)
  - Incomplete citations:
    - [E. Bengio et al., 2021] was published in NeurIPS 2021.
    - [Dai et al., 2020] was published in NeurIPS 2020.
    - [Deleu et al., 2022] was published in UAI 2022.
    - [Grathwohl et al., 2021] was published in ICML 2021.
    - [Malkin et al., 2022] will be published in NeurIPS 2022.

**Summary Of The Paper:**

This paper proposes a way to regularize GFlowNet policies using an optimal transport cost that can encourage either similarity or dissimilarity of policies at states that are close in the state space. This cost is converted into several proxy regularization terms that can be added to the trajectory balance objective in on-policy training of GFlowNets. Experiments are performed on three domains from recent work.

**Summary Of The Review:**

There is a potential for a good contribution here: optimal transport is an interesting way to answer the question mentioned in "Originality" above. However, I vote to reject because the weaknesses listed above are dominating: in particular, the lack of (1) experiments that give insight about hypothesized benefits of the prosed regularization and (2) consideration of simpler algorithms, like entropy regularization on $P_F$ or $P_B$, that may have a similar effect.

Post-rebuttal update: After reading all reviews and responses, I increase the recommendation from 3 to 5; see comments.

---

> ### Author Response · Authors · 2022-11-18
> **Author's Response to Reviewer ayzy (1)**
>
> ### Response:
>
> Thank you for your thoughtful review and valuable feedback. Below we address your concerns.
>
> **Q1**. Many of the results are unconvincing. For the hypergrid environment, the difference between algorithms is almost invisible, and it could be compensated by small changes in learning rate. I would suggest trying larger values of H and D, where sparsity may become more important, and showing more clearly the dependence on the most important hyperparameters (regularization weight -- how was it chosen?). In the EB-GFN experiments, there is practically no difference between the proposed algorithm and baselines. Furthermore, the values highlighted in Table 4 are sometimes not actually the best one in their columns, which is misleading.
>
> **Reply**:
> * First, we agree that in the hypergrid environment, the results reported are quite difficult to observe the expected behaviors of the proposed regularization method, although they exist. Then following your recommendation, we will further investigate the proposed path regularization via OT with different values of the regularization coefficient $\lambda$ in the hypergrid environment in Section 4.1. While details about the model's architecture, hyper-parameters, and evaluation criteria are the same as those provided in Appendix E.1, we will study a sparser hypergrid environment with the number of dimensions $N=8$ and the side length $H=8$ to have a better insight about the effect of the proposed regularization method and the regularization coefficient $\lambda$. In addition, the regularization coefficient is selected from the set $(0.001, 0.01, 0.1, 0.4)$. We plot the mean results over $10$ runs for each configuration in Fig. 1 here: https://sites.google.com/view/improve-gflownets-ot-rebuttal.
>   *  Note that the good range of values for the regularization coefficient is observed to highly depend on the specific setting of the experiment task. Here, we can see that when $\lambda$ is relatively small, such as $\lambda \in (0.001, 0.01)$, the performance of GFlowNets trained additionally with our proposed regularization via 0T does not seem to be significantly different from the baseline model's performance, which holds for both UB OT, Min OT, and Max OT. This may results from the small contribution of regularization to the regularized training objective, which is caused by the not large enough value of $\lambda$. Specifically, when $\lambda = 0.01$, we can still see that the performance of GFlowNets trained by minimizing the upper bound is slightly better than the baseline's result.
>   * In addition, when $\lambda$ is relatively large ($\lambda = 0.4$), the result of learned GFlowNets is even worse than the baseline, which may be due to the large value of $\lambda$ forces the model's learning focus on the regularization part more than necessary, which badly affects the optimization of the main training objective (trajectory balance objective). Specifically, this can be observed in the lower KL divergence of both UB OT, Min OT, and Max OT compared to the baseline.
>   * Meanwhile, when $\lambda = 0.1$, GFlowNets trained by minimizing the OT regularization and its upper bound clearly perform better than the baselines regarding the number of modes found and KL divergence between the actual and empirical distribution, which proves our motivation that minimizing the proposed path regularization is more beneficial in this circumstances. Otherwise, Max OT seems unsuitable because of its motivation to improve the model's exploration, while we only need the high-reward states near the corners, and the majority of states have minimal rewards.
>   * As a result, we will replace the results on the $7-D$ hypergrid environment with the $8-D$ one to demonstrate a better effect of our method.

---

> > ### Author Response · Authors · 2022-11-18
> > **Author's Response to Reviewer ayzy (2)**
> >
> > **Reply (continue to Q1)**:
> > * Second, we agree that the empirical results seem marginal in Synthetic EB-GFN tasks if we look at the improvement in values of NLL and MMD. However, the results figure out an interesting thing. We observe that the performance of training EB-GFN with Min OT and UB are quite similar. This shows that our upper bound is a good approximation when the given DAG does not satisfy the assumptions in Theorem 3.2, or when there does not exist a closed-form formulation to use Min OT.
> >   * To compute MMD and NLL values, we use 10 different sets of 4000 true samples; each set was generated by a true oracle with seeds from 1 to 10 and reported the 10 times average results. We keep the setting the same as in [1]. However, there is a gap between them. So we only use our reproduced results of EB-GFN to compare with Min OT, UB, and Max OT. That is why we do not consider the results of the EB-GFN tasks in our paper when highlighting the best results.
> >
> > **References**:
> >
> > [1]: Dinghuai Zhang, Nikolay Malkin, Zhen Liu, Alexandra Volokhova, Aaron C. Courville, and Yoshua Bengio. Generative flow networks for discrete probabilistic modeling. ICML 2022.
> >
> > ----
> >
> > **Q2**. It is interesting that the Max OT approach performs best in the active learning settings. What is the interpretation of this result? Are more entropic policies beneficial in the AL setting? This is a potential strong point that should be investigated further.
> >
> > **Reply**: We agree that maximizing the proposed path regularization via OT performs best in active learning settings. Indeed, the reasons for this good performance can be listed as follows:
> > * In the multi-round active learning settings, such as molecule generation or biological sequence design, the diversity and novelty of the generated samples become the most crucial concern of GFlowNets ([1], [2], [3]}), such that the generated candidates are expected to be good enough to survive in the later evaluation phases. Moreover, in these settings, the rewards for learning GFlowNets are also provided by a supervised approximating proxy because of limited access to the true oracle. However, because the approximating proxy is quite unreliable, diverse candidates will be helpful. In addition, diverse candidates also provide better knowledge to update the approximating proxy. Then in this circumstance, more entropic policies will be intuitively beneficial to GFlowNets for generating more diverse and novel candidates.
> > * As analyzed in Section 3.1, the upper bound of the proposed regularization contains the entropy of forward policy $\mathbf{H}(P_{F}(\cdot| s))$, then maximizing the regularization will expectedly increase the upper bound, which also will expectedly increase the entropy of the forward policy.
> >   * In addition, when the training policy is given by the forward policy $P_{F}(.| .;\theta)$, the upper bound contains the entropy of the trajectory path $\mathbf{H}(P(\tau))$, which also means that maximizing the regularization will additionally increase the diversity of the sampled trajectory.
> >   * Moreover, our proposed regularization will have an additional effect by viewing the OT cost as the minimum expectation of the transport costs. Precisely, as analyzed in Section 3.1, maximizing the regularization will increase the transport cost between each child $u_i$ of $s$ to each child $v_j$ of $s$ where $s$ and $s'$ are two neighbor states ($s\rightarrow s' \in \mathcal{A}$). And because the transportation cost is designed to be inversely proportional to the transportation probability, this effect implies that from each child $u_i$ of $s$, we will highly have a chance to move to another state that is not a child of $s$', or we will sample candidates more diversely.
> >
> > **References**:
> >
> > [1]: Moksh Jain, Emmanuel Bengio, Alex Hern ́andez-Garc ́ıa, Jarrid Rector-Brooks, Bonaventure F. P.Dossou, Chanakya Ajit Ekbote, Jie Fu, Tianyu Zhang, Michael Kilgour, Dinghuai Zhang, Lena Simine, Payel Das, and Yoshua Bengio. Biological sequence design with gflownets. ICML 2022.
> >
> > [2]: Emmanuel Bengio, Moksh Jain, Maksym Korablyov, Doina Precup, and Yoshua Bengio. Flow network based generative models for non-iterative diverse candidate generation. NeurIPS 2021.
> >
> > [3]: Nikolay Malkin, Moksh Jain, Emmanuel Bengio, Chen Sun, and Yoshua Bengio. Trajectory balance: Improved credit assignment in gflownets. NeurIPS 2022.

---

### Official Review · Reviewer_J937 · 2022-10-24

**Confidence:** 4
**Correctness:** 3
**Technical Novelty And Significance:** 4
**Empirical Novelty And Significance:** 4
**Recommendation:** 6

**Clarity, Quality, Novelty And Reproducibility:**

Overall, the paper is well written, and easy to follow. It provides all the necessary background on both GFlowNets and Optimal Transport to fully understand the paper. The work is novel.

There are some minor typos, that do not impact the clarity. I have noted a few:
 - Section 2.1, paragraph 2: "moving" -> "removing"
 - Definition 3.1: "shortage" -> "shortest"
 - Page 5 (paragraph below Fig. 1): "form" -> "from"
 - Page 5 (same paragraph): "designed" -> 'design"

The authors also provided the source code for their experiment as part of the Supplementary material, and provide all the details about the hyperparameters they chose in the Appendix. Unfortunately due to the lack of time for the review, I did not check the source code in details, nor some of the proofs in details in the Appendix.

**Details Of Ethics Concerns:**

No ethics concerns

**Strength And Weaknesses:**

**Strengths**: This regularizer is a well-founded way to improve diversity of samples from a GFlowNet. The upper-bound provides an effective strategy to add path regularization to GFlowNets. The authors evaluated this regularizer thoroughly on multiple environments over multiple metrics of interest in previous works.

**Weaknesses**:
 1. Since the core of the paper is about the added benefit of the regularizer over standard GFlowNets, I would have appreciated a more thorough study of the effect of the regularization constant $\lambda$ on the performance and diversity. What trade-offs should one expect from setting the regularization to some value? Currently, the regularization constants (provided in Appendix E) are task-specific, which is expected, but there is no mention how those regularization constants were chosen: did you choose them based on a validation set? What was the metric you optimized for?
 2. The definition of the cost function $C_{ij} = \min_{\tau} (-\log P(\tau\mid u_{i}))$ in Equation 9 seems incorrect (except for $u_{i} \equiv v_{j}$): for example you consider that if $u_{i} \not\rightarrow v_{j}$, then $C_{ij} = -\log (P_{B}(s\mid u_{i})P_{F}(s'\mid s)P_{F}(v_{j}\mid s'))$. However, we can imagine having a trajectory $\tau = u_{i} \rightarrow s \rightarrow s'' \rightarrow s' \rightarrow v_{j}$ that has a smaller $-\log P(\tau\mid u_{i})$; intuitively, it might be more unlikely to move from $s$ to $s'$ via an intermediate step $s''$, rather than directly.
 3. Related to 2., the proof of Theorem 3.2 might also be erroneous, for similar reasons. The proof in Appendix D.2 starts with $C_{ij} = -\log (P_{B}(s\mid u_{i})P_{F}(s'\mid s)P_{F}(v_{j}\mid s'))$, even though there is no assumption in Theorem 3.2 preventing us from having a situation as above with $\tau = u_{i} \rightarrow s \rightarrow s'' \rightarrow s' \rightarrow v_{j}$. All we can say, and that's what you properly used in the proof of Theorem 3.1 for the upper bound, is that $C_{ij} \leq -\log (P_{B}(s\mid u_{i})P_{F}(s'\mid s)P_{F}(v_{j}\mid s'))$.
 4. There are some confusions regarding the notions of entropy and cross-entropy. The formula in Equation 19 only corresponds to the entropy of $P(\tau)$ if $\pi_{\theta} = P$. Moreover since $P_{B}^{\star}$ is not a probability distribution, the cross-entropy $H(P_{F}(\cdot\mid s_{t}), P_{B}^{\star}(\cdot\mid s_{t}))$ is not properly defined. One can understand what this means from the Proof in Appendix D.1, but you should be precise in your use of these terms, especially since you are drawing conclusions about the behavior of the regularizer from maximizing entropies (Section 3.1) and minimizing cross-entropies (Section 3.2).
 5. The behavior of Trajectory Balance with and without OT regularization seem to be very similar on the synthetic task (Figure 2). It would be informative to add shaded areas to show the variance for the 10 runs, and evaluate if the effect of the OT regularization is statistically significant on this environment.

**Summary Of The Paper:**

This paper introduces a novel path regularizer for Generative Flow Networks (GFlowNets, Bengio et al., 2021), based on concepts from Optimal Transport (OT). This regularizer, defined in terms of directed distances in the GFlowNet, is in general expensive to compute. Fortunately, the authors show that this regularizer can be upper-bounded by a term more convenient for optimization. Moreover, under some conditions on the structure of the GFlowNet that are met in practice for the applications considered in prior works, the authors show that this regularizer can be OT regularizer can even be computed in closed form. This regularizer was tested in conjunction with the GFlowNet on a synthetic environment, discrete probabilistic modeling, and biological sequence design, and shows advantages in terms of performance and diversity compared to the baseline model without the regularizer.

---

*Yoshua Bengio, Tristan Deleu, Edward J. Hu, Salem Lahlou, Mo Tiwari, and Emmanuel Bengio. Gflownet foundations.*

**Summary Of The Review:**

This paper provides a practical solution to improve GFlowNets, and is completely novel. I am currently leaning towards acceptance, but there are some technical aspects raised here that may have significant effects on the results presented in this paper (notably Theorem 3.2). Therefore I can only recommend borderline acceptance for now.

---

> ### Author Response · Authors · 2022-11-18
> **Author's Response to Reviewer J937 (1)**
>
> ### Response:
>
> Thank you for your thoughtful review and valuable feedback. Below we address your concerns.
>
> ----
>
> **Q1**. Since the core of the paper is about the added benefit of the regularizer over standard GFlowNets, I would have appreciated a more thorough study of the effect of the regularization constant $\lambda$ the performance and diversity. What trade-offs should one expect from setting the regularization to some value? Currently, the regularization constants (provided in Appendix E) are task-specific, which is expected, but there is no mention how those regularization constants were chosen: did you choose them based on a validation set? What was the metric you optimized for?
>
> **Reply**:
> * First, to study the effect of different $\lambda$ on the performance of GFlowNets, we will further investigate the proposed path regularization via OT with different values of the regularization coefficient $\lambda$ in the hypergrid environment in Section 4.1. While details about the model's architecture, hyper-parameters, and evaluation criteria are the same as those provided in Appendix E.1, we will study a sparser hypergrid environment with the number of dimensions $N=8$ and the side length $H=8$ to have a better insight about the effect of the proposed regularization method and the regularization coefficient $\lambda$. In addition, the regularization coefficient is selected from the set $(0.001, 0.01, 0.1, 0.4)$. We plot the mean results over $10$ runs for each configuration in Fig. 1 here: https://sites.google.com/view/improve-gflownets-ot-rebuttal.
>      * Note that the good range of values for the regularization coefficient is observed to highly depend on the specific setting of the experiment task. Here, we can see that when $\lambda$ is relatively small, such as $\lambda \in (0.001, 0.01)$, the performance of GFlowNets trained additionally with our proposed regularization via 0T does not seem to be significantly different from the baseline model's performance, which holds for both UB OT, Min OT, and Max OT. This may be resulted from the small contribution of regularization to the regularized training objective, which is caused by the not large enough value of $\lambda$. Specifically, when $\lambda = 0.01$, we can still see that the performance of GFlowNets trained by minimizing the upper bound is slightly better than the baseline's result.
>      * In addition, when $\lambda$ is relatively large ($\lambda = 0.4$), the result of learned GFlowNets is even worse than the baseline, which may be due to the large value of $\lambda$ forces the model's learning focus on the regularization part more than necessary, which badly affects the optimization of the main training objective (trajectory balance objective). Specifically, this can be observed in the lower KL divergence of both UB OT, Min OT, and Max OT compared to the baseline.
>      * Meanwhile, when $\lambda = 0.1$, GFlowNets trained by minimizing the OT regularization and its upper bound clearly perform better than the baselines regarding the number of modes found and KL divergence between the actual and empirical distribution, which proves our motivation that minimizing the proposed path regularization is more beneficial in this circumstances. Otherwise, Max OT seems unsuitable because of its motivation to improve the model's exploration, while we only need the high-reward states near the corners, and the majority of states have minimal rewards.
>      * As a result, we will replace the results on the $7-D$ hypergrid environment with the $8-D$ one to demonstrate a better effect of our method.
> * Second, it is true that the regularization coefficients provided in Appendix E are task-specific. Specifically, $\lambda$ was chosen from a predefined set of values with different scales in each task, as in the previous ablation study. However, because all tasks in our experiment parts do not change the target distribution between the training and test time, $\lambda$ reported in the Appendix was chosen to have the best model's performance.

---

> > ### Author Response · Authors · 2022-11-18
> > **Author's Response to Reviewer J937 (2)**
> >
> > **Q2**. The definition of the cost function $C_{ij} = \min_{\tau} (-\log P(\tau\mid u_{i}))$ in Equation 9 seems incorrect (except for $u_{i} \equiv v_{j}$ ): for example you consider that if $u_{i} \not\rightarrow v_{j}$, then $C_{ij} = -\log (P_{B}(s\mid u_{i})P_{F}(s'\mid s)P_{F}(v_{j}\mid s'))$ . However, we can imagine having a trajectory $\tau = u_{i} \rightarrow s \rightarrow s'' \rightarrow s' \rightarrow v_{j}$ that has a smaller $-\log P(\tau\mid u_{i})$; intuitively, it might be more unlikely to move from s to s’ via an intermediate step s”, rather than directly.
> >
> > **Reply**: We agree with your opinion here because of our deficiency in not clearly stating the construction of the cost in Eqn. 9 (Eqn. 8 in revised version). Specifically, the cost matrix in Eqn. 9 (Eqn. 8 in revised version) is only an approximation to the actual distance defined in Definition 3.1 because of the following reasons:
> > * Indeed, the directed distance from a state $s$ to another state $s'$ is designed to be inversely proportional to the probability of going from $s$ to $s'$. And, because we consider two arbitrary states $s$ and $s'$, it is trivial that there does not always exist a sequence of forward transitions $\tau$ from $s$ to $s'$ such as $\tau = (s = s_{0} \rightarrow s_{1} \rightarrow ... \rightarrow s_{n} = s')$ where $s_{t} \rightarrow s_{t+1} \in \mathcal{A}$. This will result in the observation that many entries of the cost matrix may have infinite values, which can also make the OT cost infinite. Therefore, we consider a generalized notion of $\tau$ as a sequence of transitions $\tau = (s = s_{0} \rightarrow s_{1} \rightarrow ... \rightarrow s_{n} = s')$ where $s_{t} \rightarrow s_{t+1}$ can be a forward or backward transition, i.e, $\tau$ can be a back-and-forth trajectory ([1]). In fact, this sequence of transitions always exists because when we do not regard the direction of each edge, the given DAG can be considered an undirected connected graph.
> > * However, the underlying directed acyclic graph is unavailable during training progress because of the enormous number of states and edges connecting them. Then we can only have the approximation of the distance from each child state $u_{i}$ of $s$ to each child state $v_{j}$ of $s'$, or $\mathbf{C}_{ij}$ in Eqn. 9 (Eqn. 8 in revised version) is approximately computed by using the sub-graph containing $s$, $s'$, their child states, and the edges that connecting them. Specifically, rather than going directly from $u_i$ to $v_j$ if this edge exists, we can always move from $u_i$ to $v_j$ along a back-and-forth trajectory.
> >
> > Moreover, in the DAGs studied in our paper, we cannot decompose an action into others actions, or when $s’$ is a child of $s$ there does not exist $s''$ such that $s \rightarrow s'' \rightarrow s’$. Then we do not have to consider intermediate state $s''$ that $s \rightarrow s'' \rightarrow s’$.
> >
> > **References**:
> >
> > [1]: Dinghuai Zhang, Nikolay Malkin, Zhen Liu, Alexandra Volokhova, Aaron C. Courville, and Yoshua Bengio. Generative flow networks for discrete probabilistic modeling. ICML 2022.
> >
> > ----
> >
> > **Q3**. Related to Q2., the proof of Theorem 3.2 might also be erroneous, for similar reasons. The proof in Appendix D.2 starts with $C_{ij} = -\log (P_{B}(s\mid u_{i})P_{F}(s'\mid s)P_{F}(v_{j}\mid s'))$ , even though there is no assumption in Theorem 3.2 preventing us from having a situation as above with $\tau = u_{i} \rightarrow s \rightarrow s'' \rightarrow s' \rightarrow v_{j}$. All we can say, and that's what you properly used in the proof of Theorem 3.1 for the upper bound, is that $C_{ij} \leq -\log (P_{B}(s\mid u_{i})P_{F}(s'\mid s)P_{F}(v_{j}\mid s'))$.
> >
> > **Reply**:  As we have addressed your previous question, it is our deficiency in not clearly stating the construction of the cost in Eqn. 9 (Eqn. 8 in revised version). Specifically, each entry of the cost matrix $\mathbf{C}$ is actually calculated in practice by approximating as in Eqn. 9 (Eqn. 8 in revised version). In addition, we also have not stated clearly that Theorem 3.2 is derived with the additional assumption that the optimal transport cost between the forward policies of two neighbor states $\text{OT}\left ( P_F(\cdot| s),P_F(\cdot| s') \right )$ defined in Eqn. 6 (Eqn. 5 in revised version) where the cost matrix $\mathbf{C}$ is re-defined by its approximation as in Eqn. 9 (Eqn. 8 in revised version).
> >
> > Otherwise, Theorem 3.2 was also stated with the assumption that we cannot decompose an action into other actions, which was stated as "The reason is that there is not exist an action $a_k$ such that $a_k$ can decompose into others actions, i.e., $a_k = a_i+a_j$." in our paper. This results in that there does not exist an intermediate state $s''$ such as $s \rightarrow s'' \rightarrow s’$
> >
> > Then the results in Theorem 3.2 will still be true.

---

> > > ### Author Response · Authors · 2022-11-18
> > > **Author's Response to Reviewer J937 (3)**
> > >
> > > **Q4**. There are some confusions regarding the notions of entropy and cross-entropy. The formula in Equation 19 only corresponds to the entropy of $P(\tau)$ if $\pi_{\theta} = P$. Moreover since $P_{B}^{\star}$ is not a probability distribution, the cross-entropy $H(P_{F}(\cdot\mid s_{t}), P_{B}^{\star}(\cdot\mid s_{t}))$ is not properly defined. One can understand what this means from the Proof in Appendix D.1, but you should be precise in your use of these terms, especially since you are drawing conclusions about the behavior of the regularizer from maximizing entropies (Section 3.1) and minimizing cross-entropies (Section 3.2).
> > >
> > > **Reply**:
> > > * We agree with your opinion here that the formula in Eqn. 19 (Eqn. 13 in revised version) missed the condition that the training policy $\pi_{\theta}$ is given by the forward policy $P_{F}(.| .;\theta)$, which will be appended in our paper.  In addition, the conclusion about the behavior of the proposed regularization from maximizing entropies in Section 3.1 also missed that condition, which will be rewritten as: "Following Theorem 3.1, the upper bound of the optimal transport distance contains the entropy of forward policy $\mathbf{H}(P_{F}(\cdot| s))$. Moreover, when the training policy is given by the forward policy $P_{F}$, the upper bound also contains the entropy of the path $\mathbf{H}(P(\tau))$. Thus, maximizing the OT distance means maximizing the upper bound, which increases the entropy."
> > > * In addition, it is our deficiency that the "cross-entropy" $H(P_{F}(\cdot\mid s_{t}), P_{B}^{\star}(\cdot\mid s_{t}))$ is not defined correctly because $P_{B}^{\star}(\cdot\mid s_{t})$ is not a probability distribution. Specifically, we admit that our use of cross-entropy is confusing and incorrect. Indeed, the term "cross-entropy" and its notation $H(P_{F}(\cdot\mid s_{t}), P_{B}^{\star}(\cdot\mid s_{t}))$ are not really necessary for our paper. Precisely we can replace them with the original equation term $\sum_{u \in \text{Child}(s_t)}P_{F}(u| s_t)\log(P_{B}(s_t| u))$ without much change in meanings, as the conclusion about the behavior of the proposed regularization method from minimizing "cross-entropies" in Section 3.2 is derived from the term's original form.
> > >
> > > ----
> > >
> > > **Q5**. The behavior of Trajectory Balance with and without OT regularization seem to be very similar on the synthetic task (Figure 2). It would be informative to add shaded areas to show the variance for the 10 runs, and evaluate if the effect of the OT regularization is statistically significant on this environment.
> > >
> > > **Reply**:
> > > * We agree that results reported in the hypergrid environment are quite difficult to observe the expected behaviors of the proposed regularization, although they exist. Specifically, the mean results on the $4$-D and $7$-D hypergrid environment over 10 runs with the variance are shown in Fig. 2 here: https://sites.google.com/view/improve-gflownets-ot-rebuttal.
> > > * However, when running additional experiments for ablation studies, we have found that the behaviors of the proposed regularization method become more apparent when the hypergrid environment is sparser. Specifically, consider the hypergrid environment with the number of dimensions $N=8$ and the side length $H=8$, with more details provided in the ablation study presented earlier. By running the proposed regularization method with $\lambda=0.1$, we report the mean results over 10 runs in Fig. 3 here: https://sites.google.com/view/improve-gflownets-ot-rebuttal.
> > >   * We observe that the GFlowNet model trained by minimizing the path regularization via OT discovers modes faster than the baseline, which indicates its focus on finding directions leading to states with high rewards during the training progress rather than spending time exploring the environment. This also helps the model better discover the latent structures of the interested distribution and have lower KL divergence. We can also see that the upper bound is an efficient approximation in terms of complexity when using a positive regularization coefficient, whose performance is even better. As a result, we will replace the results on the $7-D$ hypergrid environment with the $8-D$ one to demonstrate a better effect of our method.
> > >
> > > ----
> > >
> > > We hope we have cleared your concerns about our work. We have also revised our manuscript according to your comments, and we would appreciate it if we can get your further feedback at your earliest convenience.

---

> > > > ### Author Response · Authors · 2022-11-24
> > > > **Response to Reviewer J937 - Any further questions on our current draft**
> > > >
> > > > We would like to thank you again for your thoughtful reviews and valuable feedback.
> > > >
> > > > We would appreciate it if you could let us know if our responses have addressed your concerns and whether you still have any other questions on the current draft and our rebuttal.
> > > >
> > > > We would be happy to do any follow-up discussion or address any additional comments.

---

### Official Review · Reviewer_MjKh · 2022-10-24

**Confidence:** 3
**Correctness:** 3
**Technical Novelty And Significance:** 3
**Empirical Novelty And Significance:** 2
**Recommendation:** 5

**Clarity, Quality, Novelty And Reproducibility:**

**Clarity**

While most of the details are described clearly, as discussed in the previous section, a lot of the choices are not motivated very clearly. There are also quite a few grammatical errors some of them listed below
- GFLowNets, Gflownets, GFlownets - in several places throughout the paper should all be GFlowNets (as stated in the abstract)
- Page 2, last paragraph line 1 - "we provide backgrounds" -> "we provide background "
- Page 5, below caption, "Remind that" -> "Recall that"

**Quality and Novelty**

The regularization scheme proposed in the paper is novel and explores an interesting direction to improve training of GFlowNets. However, the marginal empirical improvement limits the significance of the contribution.

**Reproducibility**

The authors have included code to reproduce their results and present sufficient details in the paper. The proofs are for the most part accompanied by appropriate assumptions and details.

**Strength And Weaknesses:**

**Strengths**

- The proposed regularization scheme is well formulated and described in sufficient detail. I think the optimal transport perspective fits training of GFlowNets quite well, and is an interesting direction to be explored.

**Weaknesses**

- While I believe the optimal control perspective is interesting, the particular instantiation proposed in the paper is not too clear and well motivated. In Section 3.1 the authors discuss the cases of maximizing and minimizing the path regularization. For motivating the minimization of the regularization term the authors argue that making the forward policy at neighboring states closer results in better generalization - but this point is not super clear to me (and also doesn't seem well supported by the experiments). In general choices in the method are clearly described but quite poorly motivated.
- In Section 3.3 the authors make the observation "any two neighbor states $s < s′$ do not have the same child state". I do not believe this observations holds true in the DAGs studied in the paper. For example, in the EB-GFN setup, the permutations of actions in a subtrajectory result in the same state.
- The improvement due to the regularization seems marginal at best, based on the empirical results reported in the paper. Another thing missing in the experiments is careful analysis of the effect of the regularization, i.e. how does varying $\lambda$ affect the performance. It is also not clear to me why the Max-OT is missing in the EB-GFN experiments? (Minor - in Table 3 specifically the authors report results with a precision on 10^-5 but as the oracle used in the task is a neural network that precision seems insignificant)

**Summary Of The Paper:**

GFlowNets learn stochastic policies to sample discrete objects proportionally to a non-negative reward. The paper proposes path regularization to improve exploration and generalization in GFlowNets. The authors begin by defining a notion of directed distance within the GFlowNet DAG, followed by the transport cost between forward policies of neighboring states. The authors then establish an upper-bound on this transport cost and discuss efficient implementation of the transport cost. The authors discuss two particular instantiations of the path regularization. Specifically, the authors claim that minimizing the transport cost results in faster adaptation to high reward regions, whereas maximizing the transport cost leads to improved coverage of the state space. This is accompanied by empirical results on various tasks studied in prior work on GFlowNets.

**Summary Of The Review:**

In summary, the paper explores an interesting direction of optimal transport to improve GFlowNet training. Some key details of the method are not well motivated and some observations seem to be incorrect. The empirical improvements are also marginal. I am thus leaning towards rejection but I encourage the authors to address the feedback during the discussion and rebuttal.

---

> ### Author Response · Authors · 2022-11-18
> **Author's Response to Reviewer MjKh (1)**
>
> ### Response:
>
> Thank you for your thoughtful review and valuable feedback. Below we address your concerns.
>
> ----
>
> **Q1**. While I believe the optimal control perspective is interesting, the particular instantiation proposed in the paper is not too clear and well-motivated. In Section 3.1 the authors discuss the cases of maximizing and minimizing the path regularization. For motivating the minimization of the regularization term, the authors argue that making the forward policy at neighboring states closer results in better generalization - but this point is not super clear to me (and also doesn't seem well supported by the experiments). In general choices in the method are clearly described but quite poorly motivated.
>
> **Reply**: Our motivation for the proposed regularization can be summarized as follows:
> * Generalization:
>    * In specific problems like probabilistic modeling in discrete spaces or modeling a synthetic hypergrid environment, we want to improve the generalization ability of GFlowNets. In other words, we expect the learned model can better discover the latent structure of the target distribution, i.e. (1) In the synthetic hypergrid environment, there are only states with high rewards near its corners, or (2) When modeling image datasets in discrete spaces, the modes, which have many zero entries, only concentrate on small regions of the space.
>    * In addition, the generalization ability of GFlowNets has been stated in [1]: it can guess and sample modes that haven't been visited yet, if there exist learnable patterns in the target distribution.
>   * By stating "Forward policies at nearby states along trajectories with high flow should be similar so they will prefer certain directions leading to high rewards." in Section 1 of our paper, we mean that the forward policies of two neighbor states are expected to be similar in the way that they both have the focused tendency of choosing the next action, which implicitly forces GFlowNets to find states with high rewards rather than exploring the environment. This prior is supposed to help the model discover the latent structure of the target distribution better. In addition, minimization of the regularization term can result in this property, which is further analyzed in Section 3.1.
> * Diversity and exploration:
>     * Conversely, in the multi-round active learning settings, such as molecule generation or biological sequence design, diversity and novelty of the generated samples become the most crucial characteristic of GFlowNets ([2], [3]), such that the generated candidates are expected to be good enough to survive in the later evaluation phases. Moreover, in these settings, the rewards for learning GFlowNets are also provided by a supervised approximating proxy because of limited access to the true oracle. However, because the approximating proxy is quite unreliable, diverse candidates will be helpful. In addition, diverse candidates also provide better knowledge to update the approximating proxy.
>   * In addition, the forward policy at each state $s$ also indicates a discrete probability measure whose supports are child states of $s$, and the mass at each support is the probability of moving from $s$ to it. And, the similarity between the forward policies of two neighbor states is analogous to the distance between these two discrete probability measures, which is defined by some metrics.
>   * Therefore, by stating, "to generate more diverse candidates, the prior encourages the dissimilarity of the forward policies", we expect to increase the distance between two discrete probability measures, which is supposed to make GFlowNets' generated samples more diverse. Specifically, in our proposed regularization in Section 3.1, we argue that the optimal transport cost between two discrete probability measures can be rewritten in terms of the minimum expectation of the transport costs, then maximizing the regularization can make the children states of two considered neighbor states far from each other in terms of probabilistic transition. Then maximizing the regularization will help GFlowNets generate more diverse and novel candidates.
>
> **References**:
>
> [1]: Dinghuai Zhang, Nikolay Malkin, Zhen Liu, Alexandra Volokhova, Aaron C. Courville, and Yoshua Bengio. Generative flow networks for discrete probabilistic modeling. ICML 2022.
>
> [2]: Moksh Jain, Emmanuel Bengio, Alex Hern ́andez-Garc ́ıa, Jarrid Rector-Brooks, Bonaventure F. P.Dossou, Chanakya Ajit Ekbote, Jie Fu, Tianyu Zhang, Michael Kilgour, Dinghuai Zhang, Lena Simine, Payel Das, and Yoshua Bengio. Biological sequence design with gflownets. ICML 2022.
>
> [3]: Emmanuel Bengio, Moksh Jain, Maksym Korablyov, Doina Precup, and Yoshua Bengio. Flow network based generative models for non-iterative diverse candidate generation. NeurIPS 2021.

---

> > ### Author Response · Authors · 2022-11-18
> > **Author's Response to Reviewer MjKh (2)**
> >
> > **Q2**. In Section 3.3 the authors make the observation "any two neighbor states $s<s’$ do not have the same child state". I do not believe this observations holds true in the DAGs studied in the paper. For example, in the EB-GFN setup, the permutations of actions in a subtrajectory result in the same state.
> >
> > **Reply**:  Actually, we mean that two neighbor states $s$ and $s'$ (not two arbitrary states) on each sampled trajectory, such that $s \rightarrow s' \in \mathcal{A}$, don't share any child state. In other words, there doesn't exist any state $s''$ satisfying $s \rightarrow s'' \in \mathcal{A}$ and $s' \rightarrow s'' \in \mathcal{A}$. This finding results from the property that each action can not be decomposed into the composition of others in the action space.
> >
> > Specifically, assuming that there exists such $s''$. Let the state-action pair $(s, a_i)$ leads to $s'$, $(s', a_j)$ leads to $s''$ and $(s, a_k)$ leads to $s''$. Then $a_i, a_j$ and $a_k$ must satisfy that: $a_i + a_j = a_k$, which means that action $a_k$ can be decomposed into action $a_i$ followed by action $a_j$ (contradicted).
> >
> > ----
> >
> > **Q3**. The improvement due to the regularization seems marginal at best, based on the empirical results reported in the paper. Another thing missing in the experiments is careful analysis of the effect of the regularization, i.e. how does varying $\lambda$  affect the performance. It is also not clear to me why the Max-OT is missing in the EB-GFN experiments? (Minor - in Table 3 specifically the authors report results with a precision on $10^-5$ but as the oracle used in the task is a neural network that precision seems insignificant).
> >
> > **Reply**:
> > * First, we agree that the empirical results reported only seem marginal in the hypergrid environment and discrete probabilistic modelling. However, we can see that max OT performs markedly better than the baseline in terms of the diversity and novelty of the generated candidates in active learning settings such as biological sequence design. Moreover, when doing additional experiments for ablation studies, we find that the behaviors of Min OT and UB OT become more apparent when the hypergrid environment is sparser.
> > * Second, to study the effect of different $\lambda$ on the performance of GFlowNets, we will further investigate the proposed path regularization via OT with different values of the regularization coefficient $\lambda$ in the hypergrid environment in Section 4.1. While details about the model's architecture, hyper-parameters, and evaluation criteria are the same as those provided in Appendix E.1, we will study a sparser hypergrid environment with the number of dimensions $N=8$ and the side length $H=8$ to have a better insight about the effect of the proposed regularization method and the regularization coefficient $\lambda$. In addition, the regularization coefficient is selected from the set $(0.001, 0.01, 0.1, 0.4)$. We plot the mean results over $10$ runs for each configuration in Fig. 1 here: https://sites.google.com/view/improve-gflownets-ot-rebuttal.
> >   * Note that the good range of values for the regularization coefficient is observed to highly depend on the specific setting of the experiment task. Here, we can see that when $\lambda$ is relatively small, such as $\lambda \in (0.001, 0.01)$, the performance of GFlowNets trained additionally with our proposed regularization via 0T does not seem to be significantly different from the baseline model's performance, which holds for both UB OT, Min OT, and Max OT. This may be resulted from the small contribution of regularization to the regularized training objective, which is caused by the not large enough value of $\lambda$. Specifically, when $\lambda = 0.01$, we can still see that the performance of GFlowNets trained by UB OT is slightly better than the baseline's result.
> >    * In addition, when $\lambda$ is relatively large ($\lambda = 0.4$), the result of learned GFlowNets is even worse than the baseline, which may be due to the large value of $\lambda$ forces the model's learning focus on the regularization part more than necessary, which badly affects the optimization of the main training objective (trajectory balance objective). Specifically, this can be observed in the lower KL divergence of both UB OT, Min OT, and Max OT compared to the baseline.
> >   * Meanwhile, when $\lambda = 0.1$, GFlowNets trained by Min OT and UB OT clearly perform better than the baselines regarding the number of modes found and KL divergence between the actual and empirical distribution, which proves that minimizing proposed regularization is more beneficial in these circumstances. Otherwise, Max OT seems unsuitable because of its motivation to improve the model's exploration, while we only need the high-reward states near the corners, and the majority of states have minimal rewards.
> >   * As a result, we will replace the results on the $7-D$ hypergrid environment with the $8-D$ one to demonstrate a better effect.

---

> > > ### Author Response · Authors · 2022-11-18
> > > **Author's Response to Reviewer MjKh (3)**
> > >
> > > **Reply (continue for Q3)**:
> > > * In addition, our paper has updated the Max-OT results for synthetic discrete probabilistic modeling tasks (Synthetic EB-GFN) in Table 4. When reproducing the baseline of GFlowNets, we use the same settings as in [1]. However, there is a gap between them. To highlight the best results, we will not consider the results of GFlowNets reported in the EB-GFN paper but still report them. We only take into account the reproduce results of EB-GFN when comparing with our methods (Min OT, Max OT, and UB OT).
> > >    * Training GFlowNets with either minimizing the path regularization via OT (Min OT) or the upper bound (UB OT) gains better NLL and MMD scores than the baseline and Max OT. We also observe that the performance of training EB-GFN with Min OT and UB OT are quite similar.
> > >
> > > | Metric| Method            |2spirals     |8gaussians    | circles    |moons      |pinwheel  |swissroll     | checkerboard   |
> > > |-|-|-|-|-|-|-|-|-|
> > > | NLL   | PCD               |20.094       |19.991    | 20.565    | 19.763    | 19.593   |20.172    |21.214         |
> > > |       |ALOE               |20.295    | 20.350   | 20.565    | 19.287    | 19.821   |20.160    |54.653         |
> > > |       |ALOE +             |20.062    |19.984    |20.570     | 19.743    | 19.576   | 20.170   | 21.142        |
> > > |       |EB-GFN (paper)     |20.050    |19.982    |20.546     |19.732     |19.554    |20.146    |20.696         |
> > > |       |EB-GFN             |20.0679   |19.9862   |**20.5598**   | 19.7324    |19.5735   |20.1599   |20.6839        |
> > > |       |EB-GFN + Max OT    |20.0673   |19.9857   | 20.5599   |19.7319    |19.5714   |20.1597   |20.6837        |
> > > |       |EB-GFN + UB OT     |20.0651   |**19.9854**  | 20.5600    |**19.7305**   | 19.5707   |20.1596   |20.6836        |
> > > |       |EB-GFN + Min OT    |**20.0640**  | 19.9855   |**20.5598**   | 19.7308    |**19.5699**  | **20.1595**  | **20.6831**   |
> > > | MMD   | PCD               |2.160     | 0.954    |0.188      | 0.962     | 0.505    | 1.382    |2.831          |
> > > |       |ALOE               |21.926    | 107.320  | 0.497     | 26.894    | 39.091   | 0.471    | 61.562        |
> > > |       |ALOE +             |0.149     |0.078     | 0.636     |0.516      |1.746     | 0.718    |12.138         |
> > > |       |EB-GFN (paper)     |0.583     |0.531     |0.305      |0.121      |0.492     |0.274     |1.206          |
> > > |       |EB-GFN             |0.3012    |0.0408    |-0.1724    |-0.1744    |0.2056    |0.1555    |-0.0986        |
> > > |       |EB-GFN + Max OT    |0.3258    |0.0197    |-0.1919    |-0.0456    |0.1377    |0.0763    |-0.0903        |
> > > |       |EB-GFN + UB OT     |0.2902    |**0.0102**   | **-0.2819**  | -0.1253    |0.1561    |**0.0257**   | -0.0923        |
> > > |       |EB-GFN + Min OT    |**0.1816**   | 0.0343    |-0.2775    |**-0.1966**   | **0.1220**   | 0.1334    |**-0.1071**   |
> > >
> > > **References**:
> > >
> > > [1]: Dinghuai Zhang, Nikolay Malkin, Zhen Liu, Alexandra Volokhova, Aaron C. Courville, and Yoshua Bengio. Generative flow networks for discrete probabilistic modeling. ICML 2022.
> > >
> > > ----
> > >
> > > We hope we have cleared your concerns about our work. We have also revised our manuscript according to your comments, and we would appreciate it if we can get your further feedback at your earliest convenience.

---

> > > > ### Comment · Reviewer_MjKh · 2022-11-21
> > > > **Response to Rebuttal**
> > > >
> > > > Thank you for considering my comments and making the relevant changes.
> > > >
> > > > **Q2**: Thank you for the clarification, I think I misunderstood that point!
> > > >
> > > > **Q3**: Thank you for running the additional experiments as well as the clarification on the ablation on $\lambda$ (which I believe should be in the main paper instead of the appendix). Unfortunately, I still feel the empirical results are marginal at best and considering my next point - significantly limit the impact of the method.
> > > >
> > > > **Q1**: Thank you for the explanation. Unfortunately I am still not fully convinced by the technical motivation. Like I said in my original review, I think OT seems like a natural fit for GFlowNets - as training GFNs is essentially an OT problem. However the formulation in terms of the forward policy at neighboring states seems lacking. Regardless I believe the authors have presented enough evidence and support for their claims so I will not let this affect my rating but it still seems worth mentioning.
> > > >
> > > > The authors have addressed some concerns I had but the major one still remains - weak empirical evidence for the stated claims of better generalization and diversity. I feel the work might benefit from a reconsideration of some key aspects and selection of more appropriate problem domains to showcase the method. I have increased my score but I still lean towards rejection.

---

> > > > > ### Author Response · Authors · 2022-11-22
> > > > > **Author's Response to Reviewer Mjkh**
> > > > >
> > > > > Thanks for your endorsement! We agree that more empirical results can be more beneficial in demonstrating the impact of our method. Your main concern is about “weak empirical evidence for the stated claims of better generalization and diversity”. For diversity, the GFlowNet trained by maximizing the path regularization via optimal transport (Max OT) performs markedly better than the other baselines in terms of the diversity and novelty of the generated candidates, which are two important metrics for many practical tasks, including biological sequence design tasks (AMP, TF Bind8, GFP, ...). We agree that the empirical results for the stated claims of better generalization still seem slightly unclear, i.e., in EB-GFN tasks. Therefore, we are currently running additional experiments on other domains to seek problems that can better demonstrate our method's effect of improving generalization ability.

---

### Author Response · Authors · 2022-11-18
**Summary of Revision**

Incorporating the comments and suggestions from all reviewers, besides fixing typos and notations, we have made the following main changes in the revised paper.

1. We have conducted additional experiments with the $8$-D hypergrid environment and varying $\lambda$ for the ablation study.

2. We have replaced the results on the $7-D$ hypergrid environment with the $8-D$ one in Figure 2 to demonstrate that minimizing the proposed path regularization improves the GFlowNet's generalization ability better in a sparser environment.

2. We have added Figure 3 in appendix F.1 to show the ablation study of how varying $\lambda$ affects the performance of the GFlowNet in the $8$-D hypergrid environment.

3. We have added Figure 4 in appendix F.2 to show the variance for the 10 runs on the $4$-D and $8$-D hypergrid environment.

4. We have added Max OT results for EB-GFN tasks in Table 4.

5. We have added more detailed explanations for the proposed path regularization methods' results on the hypergrid environment, discrete probabilistic modeling, and biological sequence design tasks.

6. We have rewritten the introduction in Section 1, primarily the prior constraints that we want to place on the underlying structure of the GFlowNet and the reason for using optimal transport distance instead of KL divergence to highlight our motivations.

7. We have removed the notion of cross-entropy to avoid confusion and rewritten Sections 3.1 and 3.3 for a more explicit and precise formulation of our proposed path regularization. We have also added more explicit assumptions for the results of Theorem 3.2.

8. We have revised and rewritten the background of GFlowNets in Section 2.1 and especially in Appendix B.

---

### Author Response · Authors · 2022-11-18
**General Response (1)**

Dear AC and reviewers,

Thanks for your thoughtful reviews and valuable comments, which have helped us improve the paper significantly. We are encouraged by the endorsements that: 1) The proposal of using optimal transport theory to regularize GFlowNets training is novel (Reviewer MjKh, J937, 86A8), and explores an interesting direction to improve training of GFlowNets (Reviewer MjKh, J937, ayzy); 2) The regularization was evaluated on a variety of experiments in previous works, which is combined with different baselines in the original papers (Reviewer J937, ayzy, 86A8). We have updated our submission based on the reviewers' feedback, and we have highlighted our revision in blue.

Three main concerns from the reviewers are: (1) our method is not well-motivated enough, (2) the formulation of our path regularization is not clear and rigid, which causes confusions for readers, and (3) the experimental results are not significantly improved compared with the original GFlowNet. We address these concerns here:

(1) **Motivation**: We believe there is an unclear explanation for the motivations of our path regularization via optimal transport. Please allow us to clarify this by clarifying the primary reasons by first providing why in specific tasks, we need to improve different abilities of the GFlowNet, such as generalization or exploration, which our proposed methods can provide.

* **Generalization**: In specific problems like probabilistic modeling in discrete spaces or modeling a synthetic hypergrid environment, we want to improve the generalization ability ([1],[4]) of GFlowNets. In other words, we expect the learned model can better discover the latent structure of the target distribution, i.e. (i) In the synthetic hypergrid environment, there are only states with high rewards near its corners, or (ii) When modeling image datasets in discrete spaces, the modes, which have many zero entries, only concentrate on small regions of the space. In these circumstances, we **expect that the forward policies of two neighbor states are expected to be similar in the way that they both have the focused tendency of choosing the following action, which implicitly forces GFlowNets to find states with high rewards rather than exploring the environment**, where many negligible states have small rewards. This prior can be imposed by **comparing the two forward policies, such as by using OT distance**.

* **Diversity and exploration**: In multi-round active learning settings, such as molecule generation or biological sequence design, diversity and novelty of the generated samples become the most crucial characteristic of GFlowNets ([2],[3]), such that the generated candidates are expected to be good enough to survive in the later evaluation phases. In addition, the **forward policy at each state $s$ also indicates discrete probability measure** whose supports are child states of $s$, and the mass at each support is the probability of moving from $s$ to it. Then, we **expect to increase the distance between two discrete probability measures, making GFlowNets' generated samples more diverse**. Specifically, in our proposed regularization in Section 3.1, we argue that we can rewrite the optimal transport cost between two discrete probability measures in terms of the minimum expectation of the transport costs, then **maximizing the regularization can make the children states of two considered neighbor states far from each other in terms of probabilistic transition in the sampling process** (designed cost is inversely proportional to transition probability).

* **OT distance**: Moreover, while the weakness of KL divergence is that it requires two interested distributions to share the same set of supports, OT can deal with this problem efficiently. Another reason is that the designed cost used in our OT distance can capture the given DAG’s structure and the GFlowNet’s flow, while directly using KL divergence cannot.

---

> ### Author Response · Authors · 2022-11-18
> **General Response (2)**
>
> (2) **Unclear formulation**: Indeed, we have rewritten the formulation of our proposed method. Let us clarify our approach by introducing each step of our formulation:
> * Specifically, by considering the forward policy at each state as a discrete measure whose supports are its child states, we need to determine the transportation cost from each child of $s$ to each child of $s'$ to compute the OT distance between the forward policies of two neighbor states $s$ and $s'$ ($s \rightarrow s' \in \mathcal{A}$). First, we define a directed distance between two arbitrary states $u$ and $v$ in the GFlowNet, which is then used as the transportation cost from $u$ to $v$. Specifically, it is designed to be inversely proportional to the probability of going from $u$ to $v$ via a sequence of transitions, where each transition can be forward or backward, i.e., a back-and-forth trajectory ([1]).
> * However, the underlying directed acyclic graph is unavailable during training progress because of the enormous number of states and edges connecting them. Then we can only have the approximation of the directed distance from each child state $u_{i}$ of $s$ to each child state $v_{j}$ of $s'$ by using trajectories in the sub-graph containing $s$, $s'$, their child states, and the edges that connect them. This motivates us to redefine the transportation cost by its approximation as in Eqn. 8 (in our revised version). Consequently, we can define the path regularization via optimal transport and derive its upper bound, closed form formulation in specific cases under certain conditions.
>
> (3) **Experimental Results**: We want to clarify that **in biological sequence design tasks, the GFlowNet trained by maximizing the path regularization via optimal transport (Max OT) performs markedly better than the other baselines in terms of the diversity and novelty of the generated candidates**. Moreover, by doing additional experiments with the $8-D$ hypergrid environment, we observe that the behaviors of our proposed regularization become more apparent when the environment becomes sparser (compared to the results in the $4-D$ and $7-D$ hypergrid environment). Specifically, **the GFlowNet model trained by minimizing the path regularization via OT (Min OT) or its upper bound (UB OT) discovers modes faster than the baseline and achieves lower KL error, especially when the environment is sparse**.  In Synthetic EB-GFN talks, the performance of EB-GFN trained by Min OT and UB OT is slightly better than Max OT, and the baseline. The results also show that the upper bound is a good approximation of Min OT.
>
> **References**:
>
> [1]: Dinghuai Zhang, Nikolay Malkin, Zhen Liu, Alexandra Volokhova, Aaron C. Courville, and Yoshua Bengio. Generative flow networks for discrete probabilistic modeling. ICML 2022.
>
> [2]: Moksh Jain, Emmanuel Bengio, Alex Hern ́andez-Garc ́ıa, Jarrid Rector-Brooks, Bonaventure F. P.Dossou, Chanakya Ajit Ekbote, Jie Fu, Tianyu Zhang, Michael Kilgour, Dinghuai Zhang, Lena Simine, Payel Das, and Yoshua Bengio. Biological sequence design with gflownets. ICML 2022.
>
> [3]: Emmanuel Bengio, Moksh Jain, Maksym Korablyov, Doina Precup, and Yoshua Bengio. Flow network based generative models for non-iterative diverse candidate generation. NeurIPS 2021.
>
> [4]: Nikolay Malkin, Moksh Jain, Emmanuel Bengio, Chen Sun, and Yoshua Bengio. Trajectory balance: Improved credit assignment in gflownets. NeurIPS 2022.
>
> ----
>
> We are glad to answer any further questions you have on our submission.

---

> > ### Comment · Reviewer_ayzy · 2022-11-18
> > **Broken link to new results**
> >
> > The link to new results that you shared ( https://sites.google.com/view/improve-gflownets-ot-rebuttal ) leads me to a page that informs me "this page does not exist". Is it possible to share these results in another way?

---

> > > ### Author Response · Authors · 2022-11-18
> > > **Author's Response to Reviewer ayzy**
> > >
> > > We are sorry for the inconvenience because of forgetting to set the site's public access. Could you please have a look and let us know if you have accessed it? Otherwise, the results of the ablation study in the $8$-D hypergrid environment are also provided in Figure 3 (Appendix F.1) in our revised version. In addition, our revised version shows the results with variance in the $4$-D and $8$-D hypergrid environment in Figure 4 (Appendix F.2).

---

### Author Response · Authors · 2022-11-18
**Any Questions from the Reviewers before the Deadline to Update Our Draft?**

Dear reviewers,

We would like to thank all reviewers again for your thoughtful reviews and valuable feedback. We have updated our manuscript and added new replies to your comments and questions with our latest experimental results. We have summarized the changes we made in the manuscript in the Summary of Revision below.

We would appreciate it if you could let us know if there are additional questions or concerns about our revision and rebuttal.

Best regards,

Authors

---

### Decision · Program_Chairs · 2023-01-20

**Decision:**

Reject

**Justification For Why Not Higher Score:**

The paper lacks theoretical motivation and convincing experimentation that supports the claims

**Justification For Why Not Lower Score:**

N/A

**Metareview: Summary, Strengths And Weaknesses:**

**Summary**

The paper proposes a OT regularizer for the  GFLow Net cost to promote diversity and generalization.
 In order to encourage diversity authors propose path regularization  maximizing the OT distance between two forward policies of two neighboring states. The cost in the optimal transport used is intuitively the shortest path between the states. for generalization authors propose to minimize this distance.

An upper bound is provided to the OT cost that is otherwise expensive to compute on the path for all states, and a closed form can be derived in a specific setup.

Reviewers discussed the paper with the authors  and thought that the strength of the work is in its use of OT to regularize GFlow nets that is novel  however the motivation linking the OT distance to the   generalization and diversity is lacking and there is   weak empirical evidence for the stated claims.



**Summary Of Ac-Reviewer Meeting:**

N/A